# Amphetamine disrupts dopamine axon growth in adolescence by a sex-specific mechanism in mice

Lauren M. Reynolds [1,2,7], Giovanni Hernandez[2], Del MacGowan[1,2],
Christina Popescu [1,2], Dominique Nouel[2], Santiago Cuesta [2,8], Samuel Burke[3],
Katherine E. Savell [4], Janet Zhao[2], Jose Maria Restrepo-Lozano[1,2],
Michel Giroux[2], Sonia Israel [2], Taylor Orsini[2], Susan He[2], Michael Wodzinski[2],
Radu G. Avramescu[2], Matthew Pokinko[1,2], Julia G. Epelbaum[2], Zhipeng Niu[2],
Andrea Harée Pantoja-Urbán[1,2], Louis-Éric Trudeau [3], Bryan Kolb[5],
Jeremy J. Day [4] & Cecilia Flores [2,6] ✉

Initiating drug use during adolescence increases the risk of developing addiction or other psychopathologies later in life, with long-term outcomes varying according to sex and exact timing of use. The cellular and molecular underpinnings explaining this differential sensitivity to detrimental drug effects remain unexplained. The Netrin-1/DCC guidance cue system segregates cortical and limbic dopamine pathways in adolescence. Here we show that amphetamine, by dysregulating Netrin-1/DCC signaling, triggers ectopic growth of mesolimbic dopamine axons to the prefrontal cortex, only in early-adolescent male mice, underlying a male-specific vulnerability to enduring cognitive deficits. In adolescent females, compensatory changes in Netrin-1 protect against the deleterious consequences of amphetamine on dopamine connectivity and cognitive outcomes. Netrin-1/DCC signaling functions as a molecular switch which can be differentially regulated by the same drug experience as function of an individual's sex and adolescent age, and lead to divergent long-term outcomes associated with vulnerable or resilient phenotypes.

Adolescence is an evolutionarily conserved period of life, encompassing the gradual transition from a juvenile to an adult state. While best characterized in humans[1], significant behavioral and neurobiological changes also demarcate an adolescent period in other mammals, including rodents[2–5]. The dopamine neurotransmitter system continues to mature into adulthood, in humans[6–10], non-human primates[11,12], and in rodents[4,13,14]; undergoing robust changes in connectivity and function during adolescence. Because dopamine circuitry development is highly shaped by ongoing experiences in adolescence, it is increasingly conceptualized as a "plasticity system"[15]. However, the molecular

[1]Integrated Program in Neuroscience, McGill University, Montréal, QC, Canada. [2]Douglas Mental Health University Institute, Montréal, QC, Canada. [3]CNS Research Group, Department of Pharmacology and Physiology, Department of Neurosciences, Faculty of Medicine, Université de Montréal, Montreal, QC, Canada. [4]Department of Neurobiology, University of Alabama at Birmingham, Birmingham, AL, USA. [5]Canadian Centre for Behavioural Neuroscience, University of Lethbridge, Lethbridge, AB, Canada. [6]Department of Psychiatry and Department of Neurology and Neurosurgery, McGill University, Montréal, Canada. [7]Present address: Plasticité du Cerveau CNRS UMR8249, École supérieure de physique et de chimie industrielles de la Ville de Paris (ESPCI Paris), Paris, France. [8]Present address: Department of Cell Biology and Neuroscience, Rutgers University, Piscataway, NJ, USA.
✉e-mail: cecilia.flores@mcgill.ca

mechanisms by which experiences in adolescence modify dopamine development and enduringly alter its function remain a topic of intense research.

Adolescent experiences with significant neurodevelopmental consequences range from essential/formative (e.g. social interactions with peers or conspecifics)[16–21] and enriching (e.g. targeted diet and exercise)[22–29], to deleterious (e.g. excessive stress, bullying)[30–35]. However, one of the experiences that leaves the most lasting mark on the adolescent brain is exposure to drugs of abuse[36,37], which epidemiological evidence indeed associates with a lifelong increase in the risk for addiction[38–42]. While addiction and substance use disorders were once thought to disproportionately affect men, the prevalence of drug abuse in women and in adolescent girls has dramatically increased[43,44], highlighting the urgent need to consider both sexes in clinical and preclinical research projects[45]. Earlier adolescent age of onset of drug use is a powerful predictor of addiction risk in both sexes[38,39,41], but marked sex differences in addiction trajectories also exist, with patterns of transition from recreational to compulsive drug use differing between men and women[46–49]. It is clear that not all adolescents face the same drug exposure on equal footing, and the mechanisms that explain how age and sex modulate the long-term effects of adolescent drug use need to be elucidated.

Guidance cues, widely studied in the context of embryonic growth[50–52], have emerged as key organizers of adolescent dopamine development[4,13,14]. In particular, the Netrin-1/DCC guidance cue system sculpts the structural and functional organization of the mesocorticolimbic dopamine pathway in adolescence by actively segregating the mesocortical and mesolimbic dopamine pathways at the level of the nucleus accumbens. This region functions as a choice point for dopamine axons to remain there and undergo DCC-dependent targeting processes, or to instead to continue growing to the prefrontal cortex[53,54]. Even subtle disruption to the establishment of mesocorticolimbic dopamine connectivity in adolescence, which can be induced by modifications in Netrin-1 or DCC expression, produce persistent dysregulation of prefrontal pyramidal neuronal structure and function. These enduring changes to the prefrontal cortex result in lasting impairments in impulse control[53,55], notably in action inhibition – a known index of vulnerability for addiction[56,57]. Evidence suggests that experiences in adolescence regulate Netrin-1 and DCC expression, but whether this regulation produces enduring impulse control deficits as a function of both the timing of the experience and the sex of the animal remains unknown. Here we combined molecular, anatomical, and behavioral analysis with targeted gene activation experiments in mice to show that the differential regulation of the Netrin-1/DCC guidance cue system by the same drug experience in adolescence encodes sex- and age-specific consequences on proximal dopamine development and on long-term cognitive outcomes.

## Results

### Experience with amphetamine in adolescence sex-specifically regulates Dcc expression in dopamine neurons

The guidance cue receptor DCC is highly enriched in dopamine neurons of the ventral tegmental area in both male and female rodents, with no difference between sexes in the percentage of dopamine neurons expressing Dcc mRNA (VTA; Fig. 1a, b)[58,59]. All DCC protein expression in the NAc of female mice is localized to dopamine axons (Supplementary Fig. 1a). In contrast, few or no dopamine axons in the PFC express DCC receptors (Supplementary Fig. 1b). The exact same segregation pattern of DCC expression is observed in male mice[60]. The expression of DCC protein and Dcc mRNA in the VTA decreases across postnatal development[58,61,62], and can be altered by experience at discrete time points in male animals. However, whether the effects of experience on Dcc expression are both age- and sex-specific remains to be explored. To address this question, we treated male and female mice with recreational-like doses of amphetamine (AMPH; 4 mg/kg;

which produces similar plasma levels in mice as recreational exposure of d-amphetamine in humans, including adolescents)[63] or saline during early adolescence and quantified Dcc mRNA one week later (Fig. 1c). Since AMPH in early adolescent male mice downregulated Dcc mRNA in the VTA by upregulating and its microRNA repressor miR-218 (Fig. 1b)[58], we also quantified miR-218 expression. Using sex as a biological variable and treatment as factors, the analysis revealed that AMPH in early adolescence downregulates significantly Dcc mRNA in the VTA of males, but not in females (Fig. 1d). The upregulation of miR-218 levels by AMPH in early adolescent males, which mediates the effects of AMPH on Dcc mRNA expresssion[58], is not evident in females (Fig. 1e). Furthermore, the negative relationship between Dcc and miR-218 levels in males in early adolescence (Fig. 1f)[58] is notably absent in females (Fig. 1g).

We next exposed male and female mice to the same AMPH treatment regimen, but this time during mid-adolescence (Fig. 1h), when we have previously seen an inability of AMPH to regulate DCC receptor expression in males[64]. We find that AMPH in mid-adolescence also produces a sex-specific effect, but interestingly in the opposite direction to what we observed in early adolescence: Dcc mRNA in the VTA is downregulated by AMPH in females only (Fig. 1i). In addition, there is a treatment by sex interaction in miR-218 expression (Fig. 1j), and while no relationship is apparent between Dcc mRNA and miR-218 in mid-adolescent males (Fig. 1k), these transcripts are significantly and negatively correlated in mid-adolescent females (Fig. 1l). Recent evidence indicates that the expression of homologs of Dcc and its ligand Netrin-1 drive sexual differentiation in c.elegans[65,66]. Our findings now demonstrate in mammals that a guidance cue receptor can be regulated in a sexually dimorphic manner in response to the same adolescent experience.

### Female mice are protected from the enduring effects of AMPH in mid-adolescence, despite the downregulation of Dcc in dopamine neurons

Our body of work has linked AMPH exposure during early adolescence to enduring changes in prefrontal cortex (PFC) dopamine structure and impulse control in male mice[54,63,67,68]. These effects are restricted to early adolescence and coincide with the ability of AMPH to downregulate Dcc expression[68]. Since AMPH does not downregulate Dcc expression in the VTA of female mice at this adolescent age, we hypothesized that it would not lead to enduring changes in PFC dopamine innervation or impulsivity. However, we predicted that in mid-adolescence, when AMPH does downregulate Dcc in females, there would be aberrant dopamine innervation to the PFC and impairments in inhibitory control. Female mice were therefore again exposed to AMPH or saline during early or mid-adolescence (Fig. 2a), we found that this regimen indeed produces robust conditioned place preference (Supplementary Fig. 2a, e). In adulthood, mice were randomly assigned to experiments to either stereologically assess the expanse of the dopamine innervation to the PFC, or to test behavioral inhibition using a Go-No/Go task (Fig. 2b). In line with our predictions, and in stark contrast to our findings in males[67], AMPH in early adolescent females does not increase the span of dopamine innervation in the PFC (Fig. 2c). Furthermore, while PFC dopamine disruptions produced by AMPH in early adolescence associate with enduring deficits in behavioral inhibition in male mice[68], early adolescent treatment in females does not lead to changes in the proportion of commission errors incurred in the Go-No/Go task (Fig. 2d). To obtained detailed information about individual performance across the No/Go task, we fit the proportion of commission error data of each mouse to a sigmoidal curve. Curve fitting revealed no differences in performance between treatment groups at the start of the task (upper asymptote, Fig. 2e), in the number of days it took them to begin showing performance improvement (M50, Fig. 2f), or in the proportion of commission errors made in the final trials (lower asymptote, Fig. 2g).

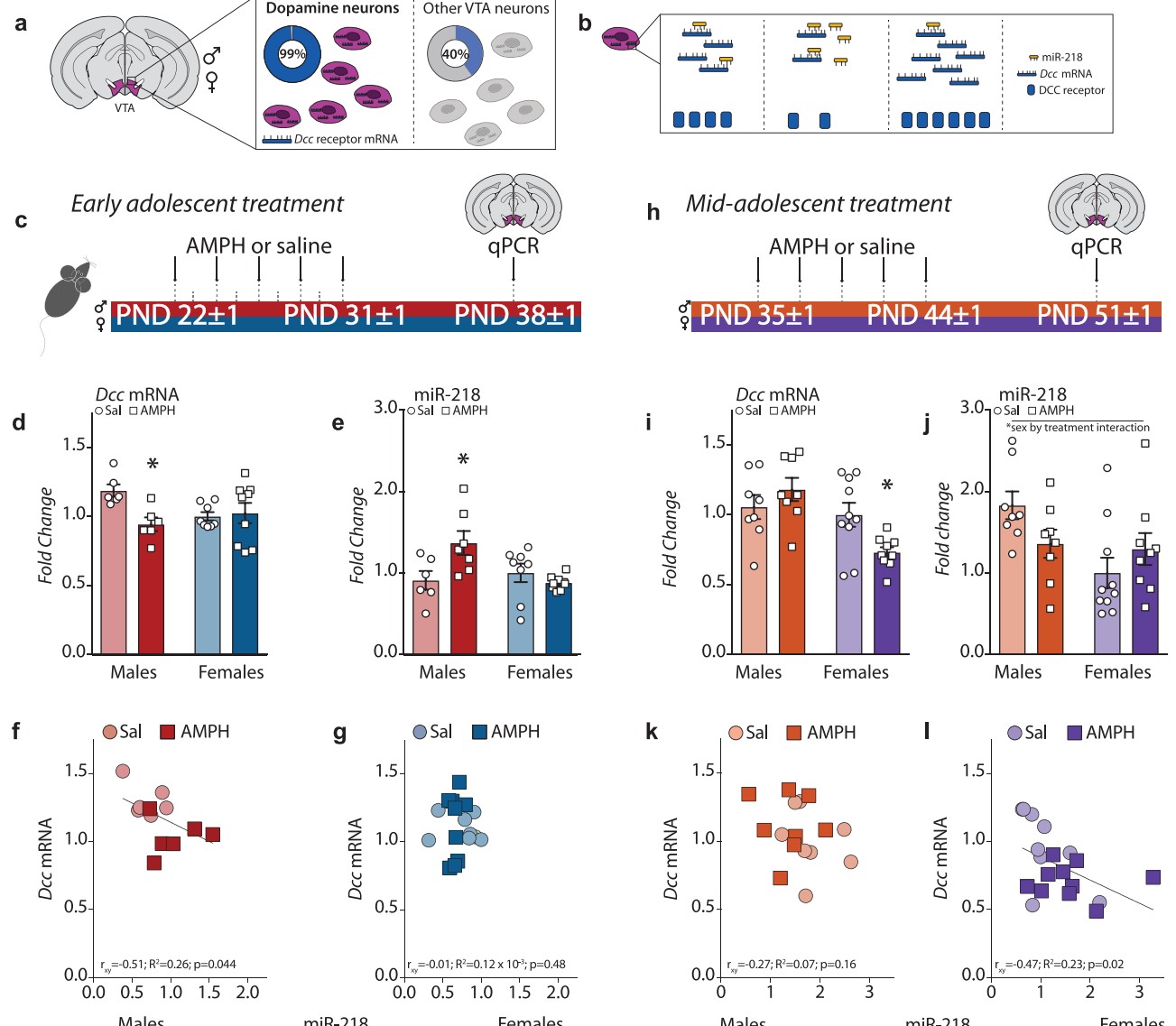

**Fig. 1 | Regulation of *Dcc* expression in the VTA by AMPH in adolescence is sexually dimorphic. a** *Dcc* mRNA is expressed by 99% of dopamine neurons in the VTA of both male and female mice[59]. **b** The microRNA miR-218 represses *Dcc* mRNA expression[58,102]. **c** Timeline of experiments in early adolescence. Male and female mice were exposed to a recreational-like amphetamine (AMPH, 4 mg/kg) regimen from P21 ± 1 to P31 ± 1[63]. One week later, *Dcc* mRNA and miR-218 expression was quantified in the VTA using qPCR. **d**–**j** AMPH in early adolescence downregulated *Dcc* expression in males, but not females (**d**) and increased miR-218 only in males (**e**) (Table 1A, B). In early adolescence, VTA miR-218 and *Dcc* mRNA levels correlated negatively in male, (**f**) but not female mice (**g**) (Table 1C, D). **h** Timeline of

experiments in mid-adolescence. Male and female mice were exposed to the same recreational-like AMPH regimen, but from P35 ± 1 to P44 ± 1 and VTA transcripts were quantified one week later. **i**–**l** In mid-adolescence, AMPH no longer altered *Dcc* mRNA in males but downregulated levels in females (**i**), and it did not significantly alter miR-218 expression in either group (**j**) (Table 1E, F). In mid-adolescence, VTA miR-218 and *Dcc* mRNA levels did not correlate in males (**k**) but were negatively correlated in females (**l**) (Table 1G, H). All bar graphs are presented as mean values ± SEM, and were normalized to the saline condition in female mice (**d**, **e**, **i**, **j**). Source data are provided as a Source Data file. *$p < 0.05$.

We also found no effect of AMPH in early adolescence on premature responses during training, a measure of waiting impulsivity (Supplementary Fig. 2b, c)[56,57], nor in correct Go responses (Hits) during the task (Supplementary Fig. 2d).

Despite the ability of AMPH to downregulate *Dcc* levels in mid-adolescent female mice (Fig. 1i), we found that exposure during this age period does not produce enduring changes in PFC dopamine innervation or in cognitive task performance (Fig. 2h, i). Indeed, when fitting individual mouse performance to a sigmoidal curve, we found no differences between treatment groups in the upper asymptote (Fig. 2j), M50 (Fig. 2k), or lower asymptote (Fig. 2l) measures. We also found no effect of AMPH in mid-adolescence on waiting impulsivity (Supplementary Fig. 2f, g), nor in correct Go responses (Hits) during

the task (Supplementary Fig. 2h). These results suggest that sex- and age-specific compensatory processes may occur in the female mouse brain to counteract the downregulation of *Dcc* mRNA by AMPH in mid-adolescence.

### *Dcc* downregulation in dopamine neurons by AMPH in mid-adolescent females is compensated by Netrin-1 upregulation in the nucleus accumbens

The primary candidate for a compensatory process in mid-adolescent females is an opposing upregulation of Netrin-1, the ligand for DCC. In male mice, Netrin-1 is expressed in the terminal regions of the meso-corticolimbic dopamine system, albeit in a complementary manner to the expression levels of DCC receptors in the dopamine axons that

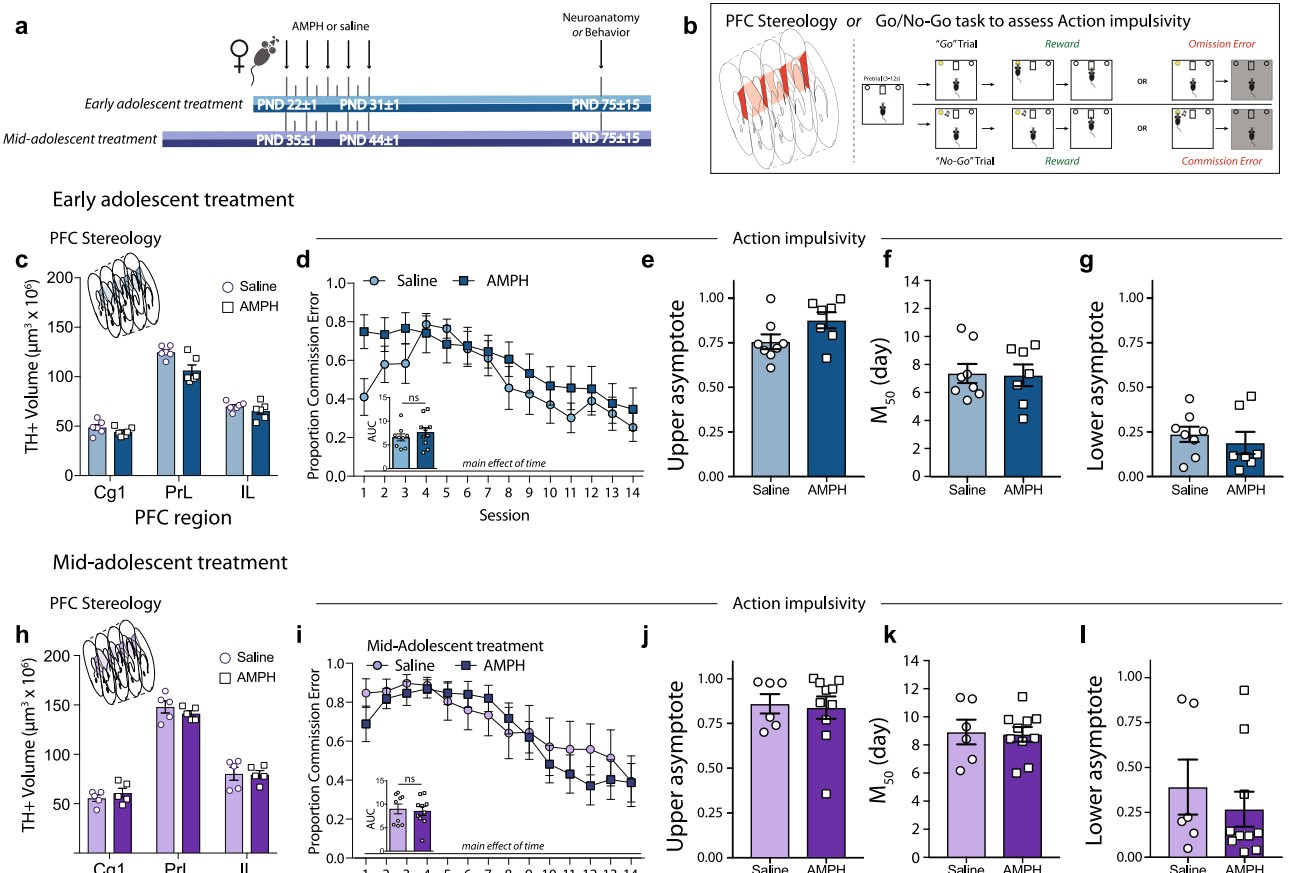

**Fig. 2 | Females are protected against the detrimental effects of AMPH in adolescence on the maturation of PFC dopamine connectivity and impulse control. a** Experimental timeline. **b** In adulthood, mice were randomly assigned to have their brains processed for stereological quantification of PFC dopamine innervation (*left schematic*) or were tested for impulse control in the Go/No-Go task (*right schematic*). **c** AMPH in early adolescence does not augment the span of the dopamine input to the cingulate (Cg1), prelimbic (PrL), and infralimbic (IL) subregions of the medial PFC in female mice, in contrast to our previous results in males[54,67]. Instead, a decrease in the volume of dopamine input to the PrL is evident (Table 2A). **d** AMPH in early adolescence does not alter action impulsivity in adulthood in female mice, unlike our previous observations in males[68]. Area under the curve (AUC) analysis indicates that the proportion of commission errors is similar between the AMPH-treated and saline groups (Table 2B, C). **e**–**g** Sigmoidal curve fit analysis (Table 2D–F) further revealed that there is no difference in the

number of commission errors at the beginning of the task (**e**, upper asymptote), that both groups began to improve their inhibitory control performance around day 7-8 (**f**, M50), and that both groups show similar proportion of commission errors during the last sessions (**g**, lower asymptote). **h** AMPH in mid-adolescence does not alter the extent of the dopamine input to the Cg1, PrL and IL subregions of the PFC in female mice (Table 2G), despite downregulating *Dcc* in dopamine neurons (Fig. 1i). **i**–**l** In mid-adolescence, females continue to be insensitive to AMPH-induced deficits in action impulsivity, with no differences in the proportion of commission errors in the task (Table 2H–I). **i** Sigmoidal curve fit analysis (Table 2J–L) revealed that AMPH and saline groups perform equally at the beginning of the task (**j**, upper asymptote), start showing improvement around day 7-8 (**j**, M50), and have similar low proportion of commission errors during the last session (**l**, lower asymptote). All graphs are presented as mean values ± SEM. Source data are provided as a Source Data file.

innervate these areas. In the PFC, Netrin-1 levels are high but only few dopamine axons express DCC. In contrast, Netrin-1 levels are lower in the nucleus accumbens (NAc), but all dopamine axons in this region highly express DCC (Fig. 3a)[60]. Netrin-1 levels in the NAc decline across adolescence in male mice[62], mirroring the same developmental pattern as we see in *Dcc* expression[58,61]. Furthermore, dopamine axons are the only source of DCC protein expression in the NAc of adult male mice[60], suggesting a crucial and complementary role of DCC in dopamine axons and Netrin-1 in the NAc in the developmental organization of mesocorticolimbic dopamine connectivity. We have recently found that this exact same pattern of DCC expression is present in adult female mice, with dopamine axons in the NAc heavily expressing DCC, and PFC dopamine axons rarely co-localizing with DCC (Supplementary Fig. 1).

We thus next assessed the effects of AMPH treatment on Netrin-1 protein expression in the NAc of the same male and female mice in which we assessed *Dcc* mRNA levels in the VTA (Fig. 3d, g; Supplementary Fig. 3). Of note, Netrin-1 is a 'sticky' guidance cue that

accumulates on the surfaces of cells[69–72], thus quantification of protein levels gives the most functionally relevant account of its properties. We observed a significant reduction in Netrin-1 protein in the NAc of males, but not females, treated with AMPH in early adolescence in comparison to saline-treated controls (Fig. 3e, f; Supplementary Fig. 3a). These findings are in line with our previous results in male mice[58], and show that the sex-specific effects of AMPH in early adolescence on *Dcc* expression in dopamine neurons extend to the regulation of Netrin-1 levels in the NAc.

Exposure to AMPH in mid-adolescence (Fig. 3g) does not alter Netrin-1 in the NAc of male mice (Fig. 3h, Supplementary Fig. 3b), consistent with their lack of sensitivity to the later timing of this drug treatment. Notably, however, AMPH in mid-adolescent females significantly *upregulates* Netrin-1 protein expression in the NAc (Fig. 3i, Supplementary Fig. 3b), indicating that at this later adolescent age, AMPH induces opposite regulation of DCC receptors in dopamine neurons and of Netrin-1 in their mesolimbic targets. In addition, we find a strong, negative correlation between *Dcc* mRNA in the VTA and Netrin-1

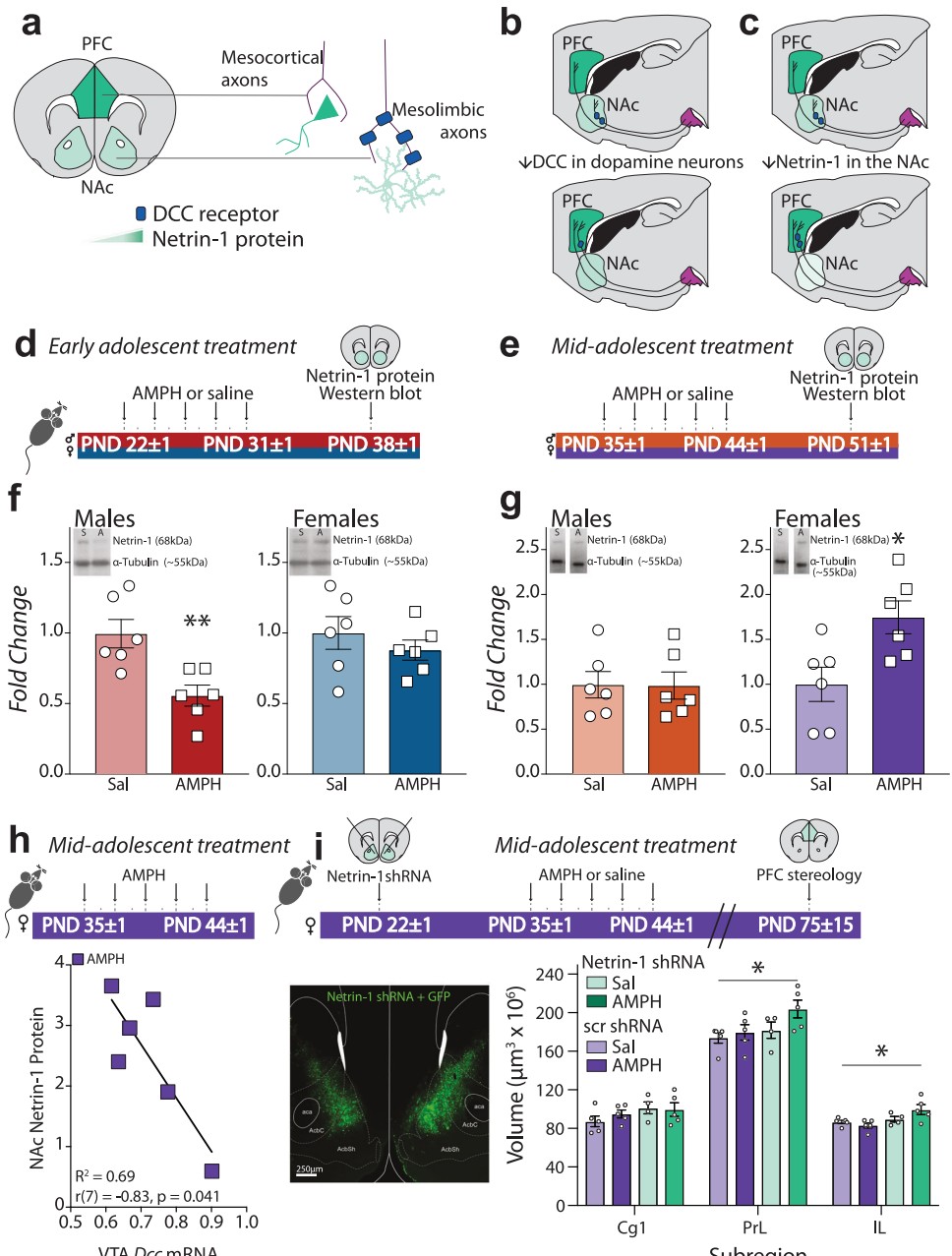

**Fig. 3 | AMPH upregulates Netrin-1 in the NAc of mid-adolescent females, counteracting its downregulation of *Dcc* levels in the VTA. a** Netrin-1 is highly expressed in the PFC, with lower expression in the NAc. DCC is expressed in a complementary pattern, with DCC-expressing dopamine axons segregated to the NAc. **b** When *Dcc* is reduced in dopamine neurons of adolescent male mice, their axons fail to recognize the NAc as their final target and instead grow ectopically to the PFC[53]. **c** Reducing Netrin-1 expression in the NAc during adolescence also results in ectopic growth of *Dcc*-expressing dopamine axons to the PFC in male mice[62]. **d** Experimental timeline for early adolescent treatment. **e** Experimental timeline for mid-adolescent treatment. **f** AMPH in early adolescence downregulates NAc Netrin-1 levels in males, but not in females (Table 3A, B). **g** AMPH in mid-adolescecence no longer downregulates Netrin-1 in the NAc, however mid-adolescent females show significant Netrin-1 upregulation in response to AMPH

(Fig. 1i) (Table 3C, D). All graphs of Western blots are normalized to the saline condition for each age and sex. **h** *Dcc* mRNA expression in the VTA and Netrin-1 protein levels in the NAc of female mice treated with AMPH in mid-adolescence show a strong and significant negative correlation (Table 3E). **i** *Top*, Netrin-1 protein in the NAc of female mice was downregulated using an shRNA approach before treatment with AMPH or saline in mid-adolescence. *Left*, Netrin-1 shRNA virus was well expressed in the NAc of female mice. *Right*, Adult female mice treated with AMPH in mid-adolescence were not different from their saline-treated counterparts when receiving a scrambled control virus, in agreement with the results in Fig. 2h. However, adult females had an increased expanse of dopamine input to the PFC as adults when Netrin-1 in the NAc was downregulated with shRNA before treatment with AMPH in mid-adolescence (Table 3F). All bar graphs are presented as mean values ± SEM. Source data are provided as a Source Data file. *$p < 0.05$, **$p < 0.01$.

protein in the NAc of mid-adolescent female mice treated with AMPH, such that mice with lower *Dcc* levels have higher levels of Netrin-1 (Fig. 3h). None of the other groups studied showed a correlation between Dcc in the VTA and Netrin-1 in the NAc (Supplementary Fig. 3). Therefore, upregulation of Netrin-1 in females may be a compensatory

effect of drug treatment, protecting against the enduring consequences triggered by drug-induced *Dcc* downregulation. To test this idea, we used an shRNA approach to downregulate *Netrin-1* expression in the NAc of female mice before subjecting them to AMPH or saline treatment in mid-adolescence (Fig. 3i). In adulthood we stereologically quantified

the expanse of the dopamine innervation to their PFC. In mid-adolescence, AMPH had no effect on the expanse of the dopamine input to the PFC in mice that received bilateral microinfusion of scrambled shRNA, in agreement with our previous results (Fig. 2h). In contrast, AMPH exposure in mid-adolescence produced an increase in the volume of the PFC dopamine input in when it was paired with *Netrin-1* downregulation in the NAc (Fig. 3i). This increase in PFC dopamine innervation volume mimics our previous results in male mice treated in early adolescence[67], showing that Netrin-1 upregulation in the NAc of mid-adolescent female mice compensates for the downregulation of *Dcc* in the VTA and protects females against the deleterious effects of AMPH on adolescent mesocorticolimbic dopamine development.

### AMPH in early adolescence induces ectopic growth of mesolimbic dopamine axons to the PFC in male mice

How exactly AMPH exposure in early adolescence produces enduring changes to PFC dopamine structure and cognitive function via *Dcc* regulation in male mice has remained a matter of debate. In contrast to the results in female mice (Fig. 2c, h), we found an increase in the span of the dopamine input to the PFC in adult males exposed to AMPH in early adolescence (Supplementary Fig. 4a), which also downregulates *Dcc* mRNA in the VTA (Fig. 1d). To determine the origin of this increase, we used intersectional viral tracing (Fig. 4a)[53] in male mice exposed to AMPH or to saline in early adolescence to track the growth of dopamine axons as they make targeting decisions at the level of the NAc. To accomplish this, we injected a retrogradely transported Cre-dependent Flp virus in the NAc of DAT[Cre] mice at PND21, while simultaneously injecting a Flp-dependent eYFP virus at the level of the VTA. This technique limits eYFP expression to dopamine neurons with terminals in the NAc at PND21. We then looked at eYFP+ dopamine axons in the PFC of adult mice, which represent axons of VTA dopamine neurons that were labeled in the NAc at the start of adolescence and which continued to grow to the PFC. We found significantly more eYFP+ dopamine axon terminals in the PFC of adult mice that were exposed to AMPH in early adolescence, in comparison to saline-exposed counterparts (Fig. 4b). This increase is in line with the overall changes in dopamine input volume seen in the same brain sections (Supplementary Fig. 4a) and previously reported[67]. The AMPH-induced increase in eYFP+ terminals in the PFC was more pronounced in ventral subregions (Fig. 4c). In addition, the number of eYFP+ dopamine terminals in the NAc is reduced in AMPH groups compared to saline controls (Fig. 4d), and there is a strong negative correlation between PFC and NAc eYFP+ dopamine terminals (Fig. 4e), indicating that AMPH in adolescence reroutes dopamine axons intended to innervate the NAc to the PFC. We find that AMPH in early adolescence also leads to significant restructuring of PFC pyramidal neuron arbors and changes in their spine density in adulthood (Supplementary Fig. 4e–i). This effect most likely results from the miswiring of dopamine axons in adolescence, as cell-autonomous manipulation of *Dcc* levels within dopamine neurons, by altering dopamine innervation to the PFC, substantially shapes the morphology of postsynaptic neurons[53,55]. Indeed, the miswiring of cortical inputs in early development has been shown to change the organization/function of local cortical networks, making them resemble those of the intended target[73]. Our results show that an experience in adolescence produces a long-distance rewiring of the developing brain, leading to enduring alterations to PFC innervation and function. We also show that this event is mediated by sex- and age- specific regulation of guidance cues.

To determine whether the ectopic growth of mesolimbic dopamine axons to the PFC is induced specifically by recreational-like doses of amphetamine (AMPH) in early adolescent male mice, we investigated whether an Adderall-like dose (0.5 mg/kg d-amphetamine, ALD), which produces similar plasma levels in mice as therapeutic treatment with d-amphetamine (trade name Adderall) in humans[63], would produce similar effects on PFC dopamine development. While

non-contingent AMPH induces a robust place preference in early adolescent male mice, the same treatment regimen with an ALD does not (Fig. 4g), nor does ALD induce a significant place preference in mid-adolescent female mice (Supplementary Fig. 4d). This is in agreement with earlier studies indicating that an ALD does not alter dopamine system function in rats[74], and does not induce long-term changes to PFC function, including deficits in inhibitory control in rodents or in non-human primates[63,75]. Notably, exposure to AMPH, but not ALD, early in adolescence decreases *Dcc* mRNA expression in the VTA of male mice[63], where 99% of dopamine neurons express *Dcc*[59]. To investigate if the rerouting effect of AMPH on NAc dopamine axons is dose-dependent, we performed the same experiments as in Fig. 4a, but comparing saline versus ALD administration (Fig. 4h). ALD in early adolescence does not alter dopamine axon growth to the PFC (Fig. 4i, j), does not produce changes in the volume of dopamine innervation (Supplementary Fig. 4b), nor does it change the number of eYFP+ dopamine terminals remaining in the NAc (Fig. 4k). It is important to note that the visual differences between the photomicrographs in Fig. 4b, i are due to sampling differences across experiments when taking images and/or to the fact the tissue used in the two experiments was processed separately. However, the basal level of eYFP+ innervation to the PFC likely does not differ between the groups, as they have similar numbers of eYFP+ varicosities in their saline conditions. Comparisons were only made within experiments where all the tissue was processed together and all the quantification was done by a single experimenter. The disruptive effects of amphetamine on dopamine development in male mice are thus linked to specific properties of recreational-like AMPH doses, which regulate the expression of *Dcc* in the VTA. While our results suggest that therapeutic-like doses of amphetamine do not impact the dopamine system of mid-adolescent female mice, a full characterization of the sex- and age-dependent effects of this treatment regimen is ongoing.

### A *Dcc*-dependent mechanism underlies the enduring deficits in impulse control induced by AMPH exposure in early adolescence

Our previous work indicates that *Dcc* expression in the VTA is important for appropriate dopamine axon targeting in adolescence, when the mesocorticolimbic dopamine pathway is still actively developing[53], suggesting DCC receptors in dopamine axons mediate the effects of AMPH effects on mesocorticolimbic dopamine development[67]. However, a causal relationship between AMPH-induced changes in *Dcc* expression and impulsivity has never been addressed, owing to limitations in tools to manipulate *Dcc* levels. While germ line and conditional knock-downs of *Dcc* expression have been investigated by our team and others, interventions to *increase Dcc* expression levels has been challenging to achieve due in part to the large size of the *Dcc* gene and mRNA. The murine *Dcc* gene spans 29 exons, contains more than 1 million base pairs[76–78], and encodes an mRNA of over 10 kilobases (NCBI reference sequence: NM_007831.3). Because of its large size, *Dcc* is not readily amenable to typical cDNA overexpression approaches. Thus, to be able to establish if *Dcc* mediates the effects of AMPH in adolescence on dopamine axon rerouting and on enduring cognitive impairments, we designed a CRISPR activation (CRISPRa) system to specifically upregulate the transcription of the *Dcc* gene in mice (Fig. 5b)[79]. Four sgRNA sequences targeting different regions -500 bp upstream of the transcription start site (TSS) of the *Dcc* gene were tested in dopamine neuron cultures, using an sgRNA targeting *LacZ* as a control (Fig. 5c). Each of the 4 single guide RNA (sgRNA) sequences tested produced a moderate increase in *Dcc* mRNA expression when compared to the *LacZ* sgRNA control (Fig. 5d). Single gene multiplexing (Fig. 5e) – i.e. combinatorial application of sgRNAs – indicates that a nearly 4-fold increase in *Dcc* mRNA could be achieved by combining the 4 sequences. For all of the following experiments, this cocktail of the 4 *Dcc* sgRNAs was used. In vivo testing (Fig. 5f)

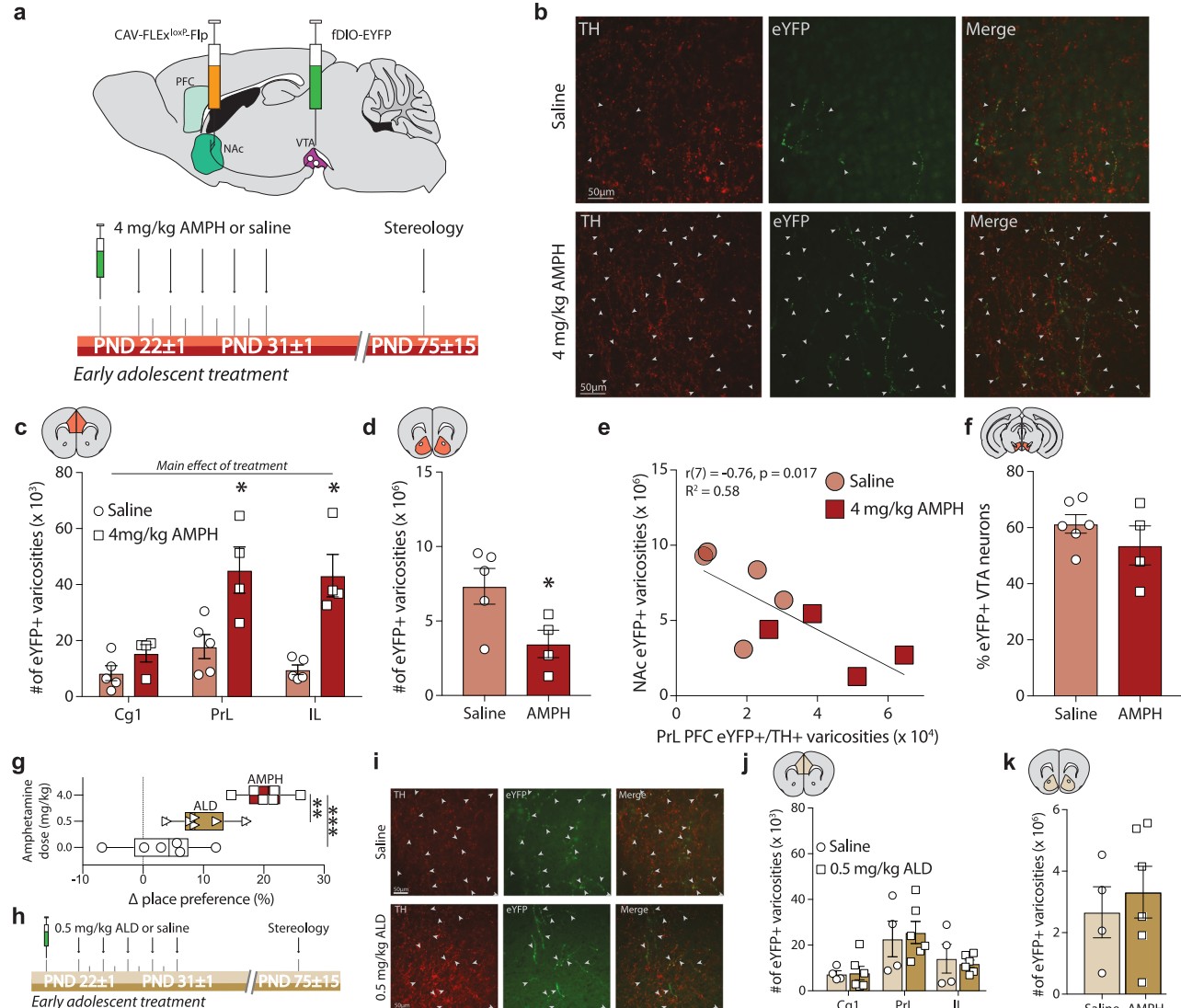

**Fig. 4 | Recreational AMPH in adolescence induces ectopic growth of meso-limbic dopamine axons to the PFC in male mice. a** Experimental design. **b** Photomicrographs showing the prelimbic PFC (PrL) of adult mice injected with tracer viruses in adolescence. *Top* Dopamine axons continued to grow from the NAc to the PFC in adolescence in the saline condition (closed arrowheads). *Bottom* The number of axons that grew to the PFC in adolescence is dramatically increased in adult mice that were exposed to AMPH early in adolescence. **c** Stereological quantification reveals a significant increase in fluorescent axon terminals across the cingulate 1 (Cg1), PrL, and infralimbic (IL) subregions of the medial PFC and a pronounced dorsal-to-ventral gradient (Table 4A), paired with a significant decrease of fluorescent terminals in the NAc (**d**, Table 4B). **e** The number of labeled terminals in the PFC and in the NAc are negatively correlated (Table 4C). **f** The percentage of VTA dopamine neuron infection is similar between treatment groups (Table 4D). **g** Exposure to AMPH induces robust conditioned place preference

(CPP) (Table 1E). This is not the case in male mice exposed to a therapeutic-like amphetamine regimen (ALD, 0.5 mg/kg). **h** Experimental design. **i** Photomicrographs showing the PrL of adult mice injected with tracer viruses in adolescence. The number of labeled axons that continued to grow from the NAc to the PFC in adolescence (closed arrowheads) is not different between adult mice that were exposed to saline (*Top*) or to ALD in adolescence (*Bottom*). ALD in adolescence does not alter the number of labeled dopamine terminals in the PFC (**j**, Table 1F) or the in the NAc (**k**, Table 1G), indicating that this AMPH dose does not interfere with dopamine axon targeting. All bar graphs are presented as mean values ± SEM. Box plots include a box extending from the 25th to 75th percentiles, with the median indicated by a line and with whiskers extending from the minima to the maxima. Source data are provided as a Source Data file. *$p < 0.05$, **$p < 0.01$, ***$p < 0.01$.

revealed that the sgRNAs can be well expressed in the VTA of adolescent male mice (Fig. 5g), and that sgRNA expression is observed in VTA dopamine neurons (Fig. 5h). Quantitative analyses revealed a significant upregulation of *Dcc* mRNA expression in the VTA (Fig. 5i), as well as a significant increase in DCC protein expression in the NAc (Fig. 5j), where DCC protein is *not* expressed by local cells – it localizes only to dopamine axons[60]. The expression of DCC protein in NAc dopamine axons is strongly correlated with mRNA expression in the VTA (Fig. 5k), indicating that the CRISPRa system produces robust upregulation of *Dcc* mRNA transcription, ultimately increasing protein translation and localization throughout the neuron.

We next asked if restoring *Dcc* expression in dopamine neurons with CRISPRa could block the effects of AMPH in early adolescence on deficits in behavioral inhibition in adulthood[68]. Male mice were injected with the sgRNA cocktail and dCas9 viruses at PND21 and then treated with a regimen of saline or AMPH (Fig. 5l). All mice were then tested in a Go-No/Go task in adulthood. All adult mice treated with saline in adolescence showed a marked reduction in commission errors across the 14 testing days, whether they received CRISPRa for *Dcc* (*Dcc* sgRNA) or the *LacZ* sgRNA control, indicative of an improvement in action inhibition across the task (Fig. 5m). Adult mice that received CRISPRa with *LacZ* sgRNA and were treated with AMPH in

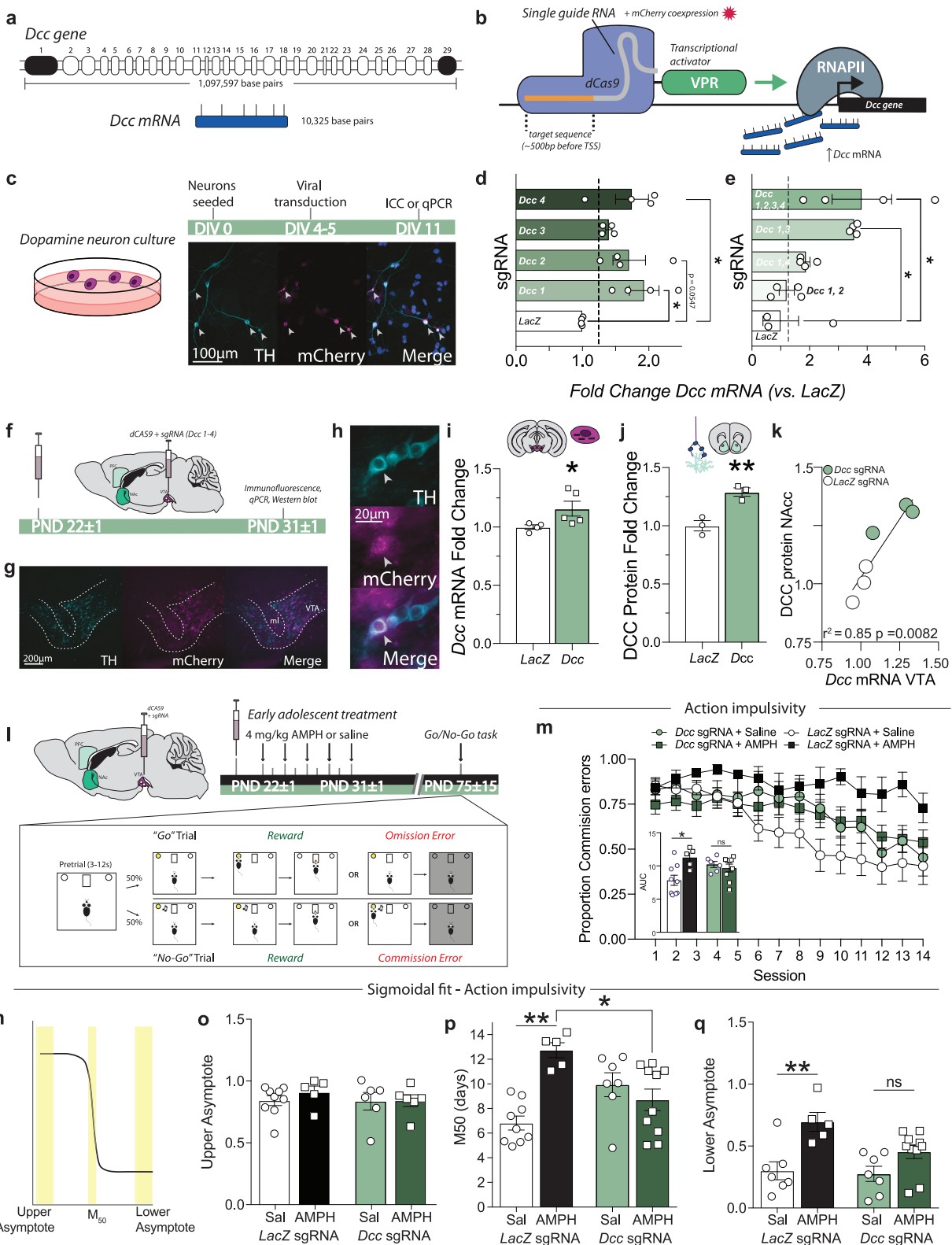

early in adolescence show deficits in behavioral inhibition, in line with previous results[68]. Strikingly, receiving CRISPRa with the *Dcc* sgRNAs prevents the development of persistent action impulsivity induced by AMPH exposure in adolescence, as this mouse group does not perform significantly different than the saline-treated groups (Fig. 5m, *inset*). While the AUC does not differ significantly between the *LacZ* and with *Dcc*-sgRNA saline groups (Fig. 5m, inset), their response curves are not visually identical. This is not surprising, considering that conditional genetic downregulation of *Dcc* expression levels in dopamine neurons produces "gene dose-dependent" effects on dopamine development, with more pronounced changes in homozygous conditional knock-outs than in heterozygotes[53,55,60]. Forthcoming studies will provide further answers to how the upregulation of *Dcc* expression impacts normative dopamine development.

To determine when in the task the differences in performance across groups emerged, individual task performance data were fitted

**Fig. 5 | CRISPRa-mediated upregulation of VTA *Dcc* transcription prevents the harmful effect of recreational AMPH in adolescence on impulse control in male mice. a** The murine *Dcc* gene and mRNA. **b** CRISPR activation (CRISPRa) system[79]. **c** Co-immunofluorescence of the mCherry tag for the *Dcc* sgRNA and TH (arrowheads) in cultured dopamine neurons (2 coverslips per guide combination). **d** All sgRNA sequences augmented *Dcc* mRNA expression (Table 5A; minimum 1.3983 ± 0.04752, maximum 1.93135 ± 0.22296 fold change). **e** The multiplex of all 4 sgRNAs gave the most robust increase in *Dcc* mRNA, with a maximal fold change of 3.81668 ± 1.03421 over *LacZ* control (Table 5B). **f** In vivo experimental design. **g** Low and **h** high magnification images of dopamine neurons (TH+) expressing the sgRNA viruses (mCherry+; arrowhead; *n* = 4 mice). **i** *Dcc* sgRNAs upregulated *Dcc* mRNA in the VTA compared to those receiving the *LacZ* sgRNA (Table 5C). **j** DCC protein expression in the NAc, where only dopamine axons express DCC receptors, was also significantly increased (Table 5D). **k** NAc DCC protein upregulation was

strongly correlated with VTA *Dcc* mRNA upregulation (Table 5E). **l** Experimental design. **m** Mice with *LacZ* sgRNA treated with AMPH in adolescence showed a greater rate of commission errors compared to mice with *LacZ* sgRNA and treated with saline. This effect of AMPH was not observed in mice that received *Dcc* sgRNA (Table 5G). Area under the curve (AUC, inset) indicates that *Dcc* CRISPRa protects against AMPH-induced action impulsivity (Table 5H). **n** Illustration of sigmoidal curve fit analysis. **o** All groups showed a similar number of commission errors at the beginning of the task (upper asymptote, Table 5I). **p** *LacZ* sgRNA AMPH-treated mice took longer to improve their task performance (M50) in comparison to the *LacZ* sgRNA saline group (Table 5J), with some mice never improving (an M50 of 14 days), an effect rescued by *Dcc* CRISPRa treatment. **q** During the last trials, only *LacZ* sgRNA AMPH-treated mice showed significant impulse control deficits (Table 2K). All bar and line graphs are presented as mean values ± SEM. Source data are provided as a Source Data file. *$p < 0.05$, **$p < 0.01$.

to a sigmoidal curve (Fig. 5n). The resulting analysis reveals that while there are no differences in initial performance (Fig. 5o, *upper asymptote*), mice that received CRISPRa with the *LacZ* sgRNA and were treated with AMPH in early adolescence take significantly longer to show improvement in the No/Go task than the other groups, evidenced by a greater $M_{50}$ (Fig. 5p). Notably, some AMPH-treated control mice do not improve at all over the task, since their $M_{50}$ value is equal to the total number of sessions. In contrast, there is no difference in the $M_{50}$ between the saline and AMPH treated mice that received CRISPRa with the *Dcc* sgRNA, indicative that CRISPRa-mediated *Dcc* upregulation protects against the enduring effect of AMPH on action inhibition. Finally, while AMPH treatment in early adolescence significantly increases the value of the lower asymptote in *LacZ* sgRNA groups, indicating profound impulse control deficits (Fig. 5q), this effect is blocked by the CRISPRa treatment targeting *Dcc* in the VTA. We also found a significant effect of AMPH in early adolescence on premature responses during training, a measure of waiting impulsivity (Supplementary Fig. 5a, b), only in the LacZ group. Waiting impulsivity and correct Go responses (Hits) during the task did not differ between the saline and AMPH treated mice that received CRISPRa with the *Dcc* sgRNA (Supplementary Fig. 5c).

## Discussion

Exposure to a drug is a necessary component to develop addiction or drug-associated psychiatric disorders, but it is not sufficient – only a subset of drug users progress to drug dependence or experience mental and behavioral disorders[80,81]. What determines why drugs have harmful consequences in some individuals, but not in others, is not well understood. Here we identify a molecular pathway that is differentially regulated by the same drug experience in adolescent mice, depending on their sex and specific age in adolescence. This different signal encodes the presence or absence of enduring negative outcomes. In early-adolescent males, but not females, a rewarding regimen of 4 mg/kg amphetamine (AMPH) downregulates *Dcc* mRNA expression in dopamine neurons. However, in mid-adolescence, AMPH downregulates *Dcc* mRNA expression in females only. Chronological age and biological sex therefore interact to modulate the impact of drugs of abuse on guidance cue receptors in adolescence. Downregulation of *Dcc* mRNA in dopamine neurons by AMPH in early adolescent males is linked to adult alterations in dopamine innervation to the PFC and to deficits in inhibitory control. Females exposed to AMPH either early or in mid-adolescence do not show these changes. AMPH in mid-adolescent females downregulates *Dcc*, but also leads to a compensatory upregulation of Netrin-1 in the nucleus accumbens, which may actively protect them against deleterious effects. We show that the male-specific deficits in PFC dopamine connectivity and cognitive function in adulthood result from AMPH-induced targeting errors by dopamine axons at the level of the developing NAc, producing incorrect segregation of mesocortical and mesolimbic dopamine projections because mesolimbic dopamine axons end up ectopically

innervating the PFC. This effect is absent upon exposure to a therapeutic regimen of amphetamine (Adderall-like dose, ALD) known not to alter *Dcc* mRNA in dopamine neurons[63], reinforcing the idea that *Dcc* downregulation negatively impacts neurodevelopment. Indeed, compensating for the downregulation of *Dcc* via CRISPRa targeted gene therapy, and therefore restoring functional DCC receptor protein levels, prevents adult cognitive impairment in males exposed to AMPH in early adolescence.

Here we show that re-routing of dopaminergic axons from the NAc to the PFC is a sex- and age-dependent consequence of an adolescent experience, namely exposure to AMPH. This may indicate a male-specific critical period in early adolescence where experiential regulation of *Dcc* and Netrin-1 can produce cortical miswiring. Understanding the mechanisms by which experiences shape the adolescent brain is still a nascent field, in contrast to well-studied early developmental periods when sensory cortices mature[82–87]. Cortical miswiring during these early critical periods profoundly shapes the target area, with its network organization more closely resembling that of the axons' intended target[73,88]. As the activity patterns and molecular profiles of mesocortical and mesolimbic dopamine axons differ markedly[89], how these rerouted connections enduringly impact PFC function and cognitive behaviors are only beginning to be understood.

Sex differences in addiction risk are well noted, with important disparities between men and women in initiation, escalation, and cessation. For example, adult women are at greater risk than men of quickly progressing to dependence shortly after initiation of cocaine use[48]. How sex differences in the enduring consequences of drug use are produced in response to the same triggering event remains poorly understood. In this study we found no evidence of a sex effect in the *immediate* rewarding effect of AMPH administration in adolescence, since all mice, regardless of age of exposure or sex, show a strong conditioned place preference for AMPH at a dose shown previously to produce peak plasma levels analogous to those seen in recreational users[63]. In contrast, we found overt sex differences in the *Dcc*-dependent neurodevelopmental impact of AMPH exposure, with females actively protected via compensatory changes in guidance cue expression. However, it is very important not to interpret this result as if females are impervious to any detrimental effects of drugs of abuse in adolescence. While females are protected against *Dcc*-dependent consequences of amphetamine exposure in early- or mid-adolescence, they may still be vulnerable to changes in other physiological systems and behavioral domains, or the effects of other drugs[90,91]. Another important point to highlight is that the vast majority of previous studies of the role of the Netrin-1/DCC system in dopamine development and in axon pathfinding were performed only in male subjects. The current study represents only a first step toward unraveling how sex influences the expression and function of this guidance cue system.

Inhibitory control is defined as the ability to suppress a given action in response to environmental cues and is known to be sensitive to changes in PFC dopamine tone[53,68,92]. Deficits in action impulsivity is

an endophenotype associated with substance abuse outcomes[56,57,93] and in adolescents this trait appears to promote the transition from recreational to compulsive drug use[94,95]. Performance on a Go/No-Go task in youth not only predicts future drug use, but also this association seems to be stronger for teens that are already heavy drug users, suggesting that action impulsivity may both predate drug use and be triggered or exacerbated by drug exposure itself[94]. Action impulsivity is a risk factor for addiction and is also considered a hallmark symptom of attention deficit/hyperactivity disorder (ADHD) which is typically managed with low dose stimulants such as amphetamine (Adderall) or methylphenidate (Ritalin)[96]. Here we show that amphetamine doses equivalent to those used therapeutically in humans do not disrupt adolescent dopamine axon targeting and growth. This is in line with results from previous research in rodents and in non-human primates[63,74,75], and with epidemiological evidence showing not only that stimulant medication itself does not increase risk for addiction, but may also counteract the enhanced predisposition for unmedicated ADHD patients to develop substance use disorders later in life[97–100].

Our findings provide important mechanistic insight regarding the critical role for the Netrin-1/DCC system in adolescent neurodevelopment[4,13], and its strong link to psychiatric disorders of an adolescent onset[101,102]. DCC receptors are not required for dopamine axons to grow to the NAc, as even mice with a homozygous deletion of DCC in dopamine neurons show dopamine innervation to the NAc[55]. As axons from dopamine neurons extend to reach anterior regions, they pass through intermediate targets along their route. The NAc is a particularly interesting structure, because it appears to be a choice point where a large number of dopamine axons establish their final connections whereas it serves merely as a waypoint for axons extending to the PFC. DCC expression in dopamine axons in adolescence determines whether dopamine axons recognize the NAc as their final target or continuing to grow to the PFC[53]. It is likely that mesocortical dopamine axons arrive early in the NAc and pause before continuing their journey to the PFC. Indeed, dopamine axons have been shown to pause at intermediate pathfinding points early in embryonic development[103–105], and guidance cues have been shown to orchestrate waiting periods in corticothalamic axon pathfinding[106]. Alternatively mesocortical dopamine axons may slowly but continuously extend from the VTA throughout adolescence.

Although the regulation of guidance cues by adolescent experience is a nascent field, recent evidence indicates that exposure to AMPH in adolescence is not the only experience that can regulate Netrin-1 and *Dcc* expression. Social defeat stress in adolescence, but not in adulthood, downregulates *Dcc* expression in dopamine neurons in male mice[35] and mild traumatic brain injury in adolescent males alters Netrin-1 in the NAc[107]. Both of these experiential regulations of Netrin-1/*Dcc* expression are associated with alterations in mesocorticolimbic dopamine circuit. Whether positive experiences could alter *Dcc* and/or Netrin-1 levels has yet to be determined but this concept has immense promise for therapeutic applications.

How susceptibility and resilience to addiction is partitioned among drug users remains largely unknown. Understanding how drug use in adolescence produces age- and sex-dependent outcomes is critical to advance addiction research, prevention, and treatment efforts. Here we show that exposure to stimulant drugs of abuse in adolescence induce axonal targeting errors, preventing the proper exclusion of mesolimbic dopamine axons from the PFC and leading to cognitive impairments that persist into adulthood. These effects are sex-specific, are mediated by the Netrin-1/DCC guidance cue system, are not observed following therapeutic-like doses, and can be prevented using gene editing strategies. We propose that Netrin-1/DCC signaling functions as a molecular switch to determine whether exposure to the same experience yields to psychiatric vulnerability or resilience.

## Methods

### Animals
Experimental procedures were performed according to the guidelines of the Canadian Council of Animal Care and approved by the McGill University/Douglas Mental Health University Institute Animal Care Committee. *DAT^Cre^* or wildtype C57BL/6J mice were bred in the Douglas Mental Health University Institute Neurophenotyping center, or were obtained from Charles River Laboratories (Saint-Constant, QC, Canada). All mice were maintained on a 12-h light–dark cycle (light on at 0800 h) in a temperature controlled (21C) facility with 42% humidity and given *ad libitum* access to food and water unless otherwise stated. Male and female mice were housed with same-sex littermates throughout the experimental procedures.

### Drugs and dose
d-Amphetamine sulfate (AMPH; Sigma-Aldrich, Dorset, United Kingdom) was dissolved in 0.9% saline. All AMPH injections were administered i.p. at a volume of 0.1 ml/10 g. A 'recreational-like' dose of 4 mg/kg was used to achieve peak plasma AMPH levels of $1300 \pm 79$ ng/mL 5 min post-injection, consistent with plasma levels induced by recreational use of AMPH in humans. A low, 'Adderall-like' dose (ALD) of 0.5 mg/kg was used to achieve peak plasma levels of $97 \pm 21$ ng/mL, in line with those observed following therapeutic administration in humans[63].

**AMPH and ALD treatment regimen.** Mice received one injection of AMPH or ALD (experimental group) or saline (control group), once every other day for a total of 5 treatment days. This treatment regimen was administered either during early adolescence (from PND $22 \pm 1$ to PND $31 \pm 1$), or during mid-adolescence (PND$35 \pm 1$ to PND $44 \pm 1$). Locomotor activity was measured 15 min prior to and 90 min after each AMPH, ALD, or saline injection.

### Axon-initiated recombination
We tracked the growth of dopamine axons across adolescence using axon-initiated recombination[53]. Importantly, we used *DAT^Cre^* mice and modified the viruses used in these experiments in order to produce cell-type specific labeling confined only to dopamine neurons. At PND21, we injected a retrogradely transported virus expressing a Cre-dependent Flp recombinase (CAV-FLEX-Flp, BioCampus Montpellier) unilaterally into the NAcc of *DAT^Cre^* mice. This design limits expression of the Flp recombinase to DAT-expressing (i.e. dopaminergic) neurons that project to the NAcc at PND 21. Simultaneously, we injected a Flp-dependent enhanced yellow fluorescent protein (eYFP) virus fDIO-eYFP (pAAV-Ef1a-fDIO-EYFP-WPRE-pA, UNC Vector Core) into the ipsilateral VTA. Thus, eYFP will only be expressed in a projection-specific and cell-type specific manner.

### Immunohistochemistry and stereology
Mice at PND$75 \pm 15$ were deeply anesthetised with a cocktail of ketamine 100 mg/kg, xylazine 10 mg/kg, acepromazine 3 mg/kg and perfused with 4% paraformaldehyde, and their brains sliced into 35μm sections using a Leica vibratome.

*For rerouting experiments in male mice*, sections were incubated for 48 h with a polyclonal anti-GFP raised in chicken (1:1000, antibody #1020, Aves labs) and a polyclonal rabbit anti-tyrosine hydroxylase (TH) antibody (1:1000, AB152, Millipore Bioscience Research Reagents). Immunostaining was visualized with Alexa Fluor 488- and Alexa Fluor 594-conjugated secondary antibodies raised in goat (1:500; Invitrogen, A11039 and A-21207).

*For DCC/TH co-labeling in adult female mice*, sections were incubated for 48 h in chicken polyclonal anti-TH (Aves lab, AB_10013440) and rabbit anti-DCC (antibody #2473, Dr. H. M. Cooper, University of Queensland, Brisbane, QLD, Australia)[108] antibodies diluted in blocking solution (1:500 dilution each) at 4 °C. Sections were then rinsed in 0.3% PBS-T and incubated in 488 goat anti-chicken and 594 donkey anti-

rabbit secondary antibodies (1:250 and 1:500 dilution; Invitrogen, A11039 and A-21207) for 2 h at room temperature. Images were taken in grayscale and channels were pseudocolored in accordance with our previous work[60].

*For TH+ stereology in female mice*, sections were incubated for 48 h with a polyclonal rabbit anti-tyrosine hydroxylase (TH) antibody (1:1000, AB152, Millipore Bioscience Research Reagents). Immunostaining was visualized with Alexa Fluor 594-conjugated secondary antibodies raised in goat (1:500; Invitrogen, A-21207).

*For the netrin-1 downregulation experiment*, sections were incubated for 48 h with a polyclonal rabbit anti-tyrosine hydroxylase (TH) antibody (1:1000, AB152, Millipore Bioscience Research Reagents). Immunostaining was visualized with an Alexa Fluor 594-conjugated secondary antibody raised in goat (1:500; Invitrogen, A-21207), the viral GFP did not require immune labeling to be clearly seen in the NAc (Fig. 3i).

Stereological procedures have been previously reported in detail[109]. Briefly, we used Stereoinvestigator (MBF, St. Albans VT) to quantify (a) the span and density of TH-positive innervation, (b) the number of TH-positive, eYFP-positive varicosities, and (c) the number of TH-negative, eYFP-positive varicosities in the nucleus accumbens (NAc) and the cingulate (Cg1), prelimbic (PrL), and infralimbic (IL) subregions of the pregenual prefrontal cortex (PFC). We also quantify (a) the number of TH-positive neurons, (b) The number of eYFP-labeled TH positive neurons, and (c) the number of TH-negative, eYFP-positive neurons in the ventral tegmental area (VTA) and substantia nigra pars compacta (SNc).

Innervation volume in cubic micrometers was assessed with the Cavalieri method in Stereoinvestigator. Cells and terminals were quantified using the optical fractionator probe. As in all our previous neuroanatomical studies, we obtained counts only from the right hemisphere because of the lateralization of dopamine systems and the unilateral injections for axon tracing experiments. The Coefficient of error for TH-positive varicosities/cells was below 0.10 for all regions of interest. Counts were performed blind.

**PFC.** Cg1, PrL, and IL subregions of the PFC were delineated according to plates spanning 14–18 of the mouse brain atlas[110] and contours were traced at 5X magnification using a Leica DM400B microscope along the dense TH-positive innervation of PFC layers V-VI. An unbiased counting frame (50 ×50 μm) was superimposed on each contour and counts were made at regular predetermined intervals ($x = 175$ μm, $y = 175$ μm) from a random start point. TH positive and eYFP positive varicosities were counted at 100X magnification on 5 sections contained within the rostrocaudal borders of our region of interest (Plates 14–18; 1:4 series). A guard zone of 4 μm was used and the optical disector height was set to 10 μm.

**NAc.** An unbiased counting frame (10 × 10 μm) was superimposed on the contour of the NAc and counts were made at regular predetermined intervals ($x = 400$ μm, $y = 400$ μm) from a random start point. Counting was performed at 100X magnification on four of the eight sections contained within the rostrocaudal borders of our region of interest (Plates 15–18, 1:4 series). A guard zone of 4 μm was used and the optical disector height was set to 5μm.

**Midbrain analysis.** The counting scheme used a 60 × 60 μm counting frame ($x = 150$ μm, $y = 150$ μm intervals) with a random start point. Counting was performed at 40X magnification in a 1:4 series. A 3 μm guard zone and a probe depth of 10 μm were used. Stereological counts of eYFP and TH co-labeled neuron populations were expressed as proportions.

**CRISPR activation**

**CRISPR/dCas9 and sgRNA construct design.** Single guide RNAs compatible with CRISPR activation (CRISPRa) were designed as previously described[79]. Briefly, sgRNA targets were designed using online tools provided by the Zhang Lab at MIT (crispr.mit.edu) and CHOPCHOP (RRID:SCR_015723; http://chopchop.cbu.uib.no/)[111,112] to target within −1000/ −500 bp of the transcription start site (TSS) of the mouse *Dcc* gene. To ensure specificity, all CRISPR RNA (crRNA) sequences were then analyzed with National Center for Biotechnology Information's (NCBI) Basic Local Alignment Search Tool (BLAST). A list of the target sequences is provided in Supplementary Table 1. Custom crRNAs were ordered as oligonucleotide sequences (Sigma Aldrich) with 5′ 4-bp overhangs (CACC for the sense strand, AAAC for the antisense strand). crRNAs were annealed, phosphorylated with PNK (NEB), and ligated using T4 ligase (NEB) into the short guide RNA (sgRNA) scaffold using the BbsI cut sites with unique overhangs mentioned above. For crRNA sequences that did not begin with a guanine, the first base of the crRNA sequence was substituted to guanine to maintain compatibility with the U6 promoter. CRISPRa experiments used lentivirus compatible plasmid constructs previously optimized for robust neuronal expression (lenti SYN-FLAG-dCas9-VPR, RRID:Addgene_114196; lenti U6-sgRNA/EF1a-mCherry, RRID:Addgene_114199)[79]. The bacterial *LacZ* gene target was used as a sgRNA non-targeting control[113].

**Lentivirus preparation**

**Plasmid preparation.** One Shot Stbl3 Chemically Competent E. coli (Invitrogen, Catalog number: C737303), were heat shock transformed to amplify all plasmids. Plasmids were purified using a Qiagen Endo-Free Plasmid Maxi Kit (Catalog number: 12362).

**Lentivirus production.** Viruses were produced in a sterile environment subject to BSL-2 safety by transfecting HEK293T cells (ATCC, catalog number: CRL-3216) with specified CRISPR-dCas9 plasmids, the psPAX2 packaging plasmid, and the pCMV-VSV G envelope plasmid (Addgene plasmids #12260 and #8454) with FuGene HD (Promega) for 48 h. Cells were incubated at 37 °C and 5% $CO_2$ in supplemented Ultraculture media (L-glutamine, sodium pyruvate, and sodium bicarbonate) in either a T75 or T225 culture flask. Viruses were purified from the supernatant using filter (0.45 mm) and ultracentrifugation (25,000 rpm, 75465 g, 1 h 45 min at 4 °C). Viral titer was determined using a qPCR Lentivirus Titration kit (Lenti-X, qRT-PCR Titration kit, Takara). After 40–48 h, lentiviruses were concentrated with Lenti-X concentrator (Takara), resuspended in sterile PBS, and used or frozen at −80 °C immediately. Only viruses with a titer of >1 ×10^15 GC/ml were used. Viruses were stored in sterile PBS at 80 °C in single-use aliquots.

**In vitro validation.** Primary mesencephalic neuron cultures were prepared from dissections of male and female postnatal day 0-2 (P0 to P2) C57/BL6J mice according to a protocol described previously[114]. Briefly, mice were cryoanesthetized and the brain was rapidly obtained to isolate the VTA and SNc. The tissue was digested with papain and triturated to obtain a single-cell suspension. The cells were plated on 15 mm diameter glass coverslips at 120 000 cells/ml on top of a pre-established cortical astrocyte layer.

For immunofluorescent imaging, neuronal cultures were fixed with 4% paraformaldehyde (PFA), permeabilized, and non-specific binding sites were blocked using BSA. Dopamine neurons were identified by immunofluorescence using a primary anti-TH antibody (1:1000, Millipore Sigma, cat. no. MAB318). Cultures were washed with PBS and incubated for 2 h at room temperature with a secondary antibody (anti-mouse Alexa Fluor-488, 1:1000, Invitrogen A-11001). Expression of the virally expressed sgRNAs and dCAS9 was validated by detecting the associated co-expressed mCherry protein with a rabbit anti-RFP (Rockland, cat. no. 600-401-379) and anti-rabbit Alexa Fluor 594- conjugated secondary antibody (1:500; Invitrogen cat. no. A21207). Finally,

coverslips were washed, counterstained with DAPI (blue) and mounted in Fluoromount-G (Southern Biotech) on Superfrost/Plus microscope slides. For qPCR experiments, mRNA was extracted with trizol from the cells on the cover slips.

**In vivo validation.** Early adolescent male mice were bilaterally infused with 1.0 µl of total lentivirus mix with 0.33 µl of the 4 sgRNAs and 0.66 µl of the dCas9-VPR virus in sterile PBS[79]. Viral transduction and *Dcc* mRNA overexpression were assessed 10 days later via immunofluorescence and qPCR in the VTA. DCC protein expression was assessed by western blot analysis in the NAc, where DCC protein is only expressed in dopamine axons[60].

**Experimental design.** Male mice received VTA stereotaxic bilateral infusions of the dCas9-VPR and *Dcc* targeting sgRNAs or (control) dCas9-VPR and *LacZ* sgRNA lentiviral constructs at P21. Two days later, they began the AMPH or saline treatment regimen. In adulthood, mice were tested in the Go/No-Go task.

**Infection and probe placement verification.** Adult mice received an overdose of ketamine (100 mg/kg), xylazine (5 mg/kg), and acepromazine (1 mg/kg) through intraperitoneal injection and were perfused intracardially with ice-cold phosphate-buffered saline (PBS, 1x) followed by ice-cold 4% paraformaldehyde (PFA, pH = 7.4). Brains were dissected, post-fixed in 4% PFA overnight at 4 °C, and transferred to 1x PBS 24 h before slicing. 35µm coronal sections were obtained using a vibratome (Leica Biosystems VT1000S) and stored in a cryo-protective solution at −20 °C until processing.

Every second section was processed for visualization of TH+ neurons and mCherry. Sections were rinsed three times for 10 min with 1x PBS and blocked in 2% bovine serum albumin (in 1x PBS and Tween-20) for 1 h at room temperature. Sections were then incubated in primary antibodies, including mouse anti-TH (Millipore Sigma, cat. no. MAB318) and rabbit anti-RFP (Rockland, cat. no. 600-401-379), for 48 h at 4 °C. Sections were rinsed three times for 10 min with 1x PBS and incubated in secondary antibodies, including goat anti-mouse Alexa Fluor 488 (Invitrogen, cat. no. A-11001) and donkey anti-rabbit Alexa Fluor 594 (Invitrogen, cat. no. A-21207), for 1 h at room temperature. Sections were rinsed three times in 1x PBS and mounted with VECTASHIELD Hardset antifade mounting medium with DAPI (Vector Laboratories, cat. no. H-1500-10). Representative images were taken using the Stereo Investigator software (MBF Bioscience) with an epifluorescent microscope (Leica DM400X3).

### Morphological analysis of PFC pyramidal neurons
**Golgi–Cox staining.** Mice were deeply anesthetized with sodium pentobarbital (>75 mg/kg; i.p.) and perfused with 0.9% saline, their brains were then processed for Golgi–Cox staining as previously[53,60,115].

**Anatomical analysis.** Basilar dendritic arbors and spines of layer V mPFC pyramidal neurons were analyzed to quantify the total arbor length, number of branches, and spine density of each cell. Neurons from the Cg1, PrL, and IL subregions of the pregenual PFC were analyzed. A Leica model DM400 microscope equipped with a Ludl XYZ motorized stage was used to identify cells, trace dendritic arbors, and quantify dendritic spines. Relevant regions were first identified at low magnification (5X objective). Cells that were chosen for tracing and analysis were required to have intact branches, well impregnated staining, and not obscured by blood vessels, astrocytes, or heavy clusters of dendrites from other cells. Neurolucida software (MicroBrightField) was used to trace the dendritic arbors of selected cells and to quantify dendritic arbor length, dendrite number, and the spine density on selected dendrite segments. For both dendritic arbor and spine density analysis, the same neurons were sampled. One dendritic segment (third-order tip or greater) was analyzed per neuron under the 100X objective. Spines were always counted from the last branch point to the terminal tip of the dendrite. No attempt was made to correct for the fact that some spines are obscured from view, so the measure of spine density necessarily underestimates total spine density. Anatomical analysis was conducted blind to treatment condition. A minimum of four cells were analyzed per brain, and averaged across each subject.

### Behavior
**Go/No-Go.** The Go/No-Go task was performed as previously[53]. Briefly, food-restricted mice (85% free feeding weight) were trained to nosepoke for chocolate-flavored dustless precision pellets (BioServ, Inc., Flemington, NJ, USA) in operant boxes (MedAssociates St. Albans, VT USA). Data was recorded using MedPCIV software (MedAssociates, St. Albans, VT USA). The mice first undergo discrimination training, where they learn to nose poke only when signaled to do so. Premature responses in discrimination training sessions are a measure of waiting impulsivity. After training, mice underwent daily sessions of the Go/No-Go Task, which requires the mice to respond to a lighted 'Go' cue or inhibit their response to this cue when presented in tandem with an auditory 'No-Go' cue. Within each session, the number of 'Go' and 'No-Go' trials were given in an approximately 1:1 ratio and presented in a randomized order. Each session lasted 30 min and consisted of approximately 30–50 'Go' and 30–50 'No-Go' trials. Number of responses to the No-Go cue (commission errors) and correct responses to the Go cue (hits) were analyzed. Commission errors represent a measure of action impulsivity, defined as a failure to appropriately inhibit behavior.

**Conditioned place preference.** Male or female mice were tested for conditioned place preference to 0.5 or 4 mg/kg AMPH in early or mid-adolescence. On day 1 mice were allowed to freely explore the CPP apparatus for 30 min, which consisted of 2 distinct chambers (one striped, one polka-dotted) and a neutral (gray) area connecting the chambers. Time spent in each chamber was measured to determine a preference percentage between the chambers for each individual animal, and a biased design was used, i.e. the less preferred chamber during the pretest would be paired with AMPH (experimental group), or with saline (control group). Following the pretest day, animals were exclusively exposed to one chamber, paired either with an AMPH (experimental group) or saline (control group) injection, for 30 min every other day for 9 treatment days. This is identical to the treatment regimen used in all anatomical, behavioral, and neurochemical experiments in this study. After the last day of injections, mice were once again allowed to freely explore the full enclosure for a 20 min post-test while the time spent in each was measured. A delta preference score was then calculated for each mouse by subtracting the percentage of time spent in the originally unpreferred chamber during the pretest from the percentage of time spent in that same chamber during the post-test, *Δ Place Preference = % time POST − % time PRE*.

### Quantitative real-time PCR
qPCR experiments were performed as previously described[58]. PND21 and 35 male and female C57BL/6J mice were rapidly decapitated and their brains were flash frozen in 2-methylbutane (Fisher Scientific, Hampton, NH, USA). Brains were sliced in 1-mm-thick coronal slices using a cryostat and VTA, NAcc, and mPFC punches were taken from the resulting sections. Total RNA and microRNA were extracted from the VTA punches using an mRNAeasy Micro Kit (Qiagen). *Dcc* mRNA was reverse transcribed using a High-Capacity cDNA Reverse Transcription Kit (Applied Biosystems), and real-time PCR was performed using a TaqMan assay kit (Applied Biosystems) on a 7900HT RT PCR system (Applied Biosystems) in technical triplicates. *Gapdh* was used as a reference gene to control for experimental variability. A TaqMan MicroRNA Reverse Transcription Kit was used alongside the

**Table 1 | Detailed statistics for Fig. 1**

| | Statistical test | Factor | n | Statistic | 95% confidence interval | p value (adjusted where appropriate) | Corresponding figure |
|---|---|---|---|---|---|---|---|
| A | | | Male Saline = 6<br>Male AMPH = 6<br>Female Saline = 8<br>Female AMPH = 9 | | | | Fig. 1d |
| | Two-way ANOVA | Interaction | | F (1, 25) = 5.452 | | **0.0279** | |
| | | Sex | | F (1, 25) = 0.8604 | | 0.3625 | |
| | | Treatment | | F (1, 25) = 3.695 | | *0.066* | |
| | Sidak's multiple comparisons test | Saline vs. AMPH within Male | | t(25) = 2.782 | 0.03866 to 0.4946 | **0.0201** | |
| | Sidak's multiple comparisons test | Saline vs. AMPH within Female | | t(25) = 0.3204 | −0.2177 to 0.166 | 0.9382 | |
| B | | | Male Saline = 6<br>Male AMPH = 6<br>Female Saline = 8<br>Female AMPH = 9 | | | | Fig. 1e |
| | Two-way ANOVA | Interaction | | F (1, 25) = 7.551 | | **0.011** | |
| | | Sex | | F (1, 25) = 3.618 | | *0.0687* | |
| | | Treatment | | F (1, 25) = 2.595 | | 0.1197 | |
| | Sidak's multiple comparisons test | Saline vs. AMPH within Male | | t(25) = 2.849 | −0.6597 to −0.05928 | **0.0172** | |
| | Sidak's multiple comparisons test | Saline vs. AMPH within Female | | t(25) = 0.8827 | −0.1589 to 0.3464 | 0.6227 | |
| C | | | Male Saline = 6<br>Male AMPH = 6 | | | | Fig. 1f |
| | Pearson r | Dcc and miR-218 | | r = −0.5129, r2 = 0.2631 | −0.8397 to 0.08641 | **0.0441** | |
| D | | | Female Saline = 8<br>Female AMPH = 9 | | | | Fig. 1g |
| | Pearson r | Dcc and miR-218 | | r = −0.01114, r2 = 0.000124 | −0.4892 to 0.472 | 0.4831 | |
| E | | | Male Saline = 8<br>Male AMPH = 8<br>Female Saline = 9<br>Female AMPH = 10 | | | | Fig. 1i |
| | Two-way ANOVA | Interaction | | F (1, 31) = 6.212 | | **0.0182** | |
| | | Sex | | F (1, 31) = 9.524 | | **0.0042** | |
| | | Treatment | | F (1, 31) = 0.7422 | | 0.3956 | |
| | Sidak's multiple comparisons test | Saline vs. AMPH within Male | | t(31) = 1.107 | −0.3724 to 0.1339<br>0.01271 to 0.478 | 0.4768 | |
| | Sidak's multiple comparisons test | Saline vs. AMPH within Female | | t(31) = 2.478 | | **0.0373** | |
| F | | | Male Saline = 8<br>Male AMPH = 8<br>Female Saline = 9<br>Female AMPH = 10 | | | | Fig. 1j |
| | Two-way ANOVA | Interaction | | F (1, 31) = 6.602 | | **0.0152** | |
| | | Sex | | F (1, 31) = 1.774 | | 0.1925 | |
| | | Treatment | | F (1, 31) = 0.01128 | | 0.9161 | |
| | Sidak's multiple comparisons test | Saline vs. AMPH within Male | | t(31) = 1.673 | −0.1911 to 1.135 | 0.198 | |
| | Sidak's multiple comparisons test | Saline vs. AMPH within Female | | t(31) = 1.977 | −1.122 to 0.09665 | 0.1107 | |
| G | | | Male Saline = 8<br>Male AMPH = 8 | | | | Fig. 1k |
| | Pearson r | Dcc and miR-218 | | r = −0.27, r2 = 0.07289 | −0.6753 to 0.2606 | 0.1559 | |
| H | | | Female Saline = 9<br>Female AMPH = 10 | | | | Fig. 1l |
| | Pearson r | Dcc and miR-218 | | r = −0.4763, r2 = 0.2268 | −0.765 to −0.02813 | **0.0196** | |

Significant p values are noted in bold text.

corresponding miRNA TaqMan probes (Applied Biosystems, Foster City, CA) to reverse transcribe and perform Real-Time PCR for miR-218, and expression levels were calculated using the AQ standard curve method. The small nucleolar RNA (snoRNA) RNU6B was used as an endogenous control to normalize miR-218 expression.

### Western blot
PND21 and 35 male and female C57BL/6J mice were rapidly decapitated and their brains were flash frozen in 2-methylbutane.

Bilateral punches from the NAc were processed for western blot as before[53,62,116]. Briefly, protein samples (20 µg) were separated on a 10% SDS-PAGE and transferred to a PVDF membrane which was incubated overnight at 4 °C with antibodies against Netrin-1 (1:1000, Abcam Inc, Toronto, ON, Canada) and a-Tubulin (1:20000, Cell Signaling, Danvers, MA, USA) for loading control. Protein bands were detected by chemiluminescence (Bio-Rad, Mississauga, ON, Canada) and analyzed using Image Lab system software (Bio-Rad, Mississauga, ON, Canada).

**Table 2 | Detailed statistics for Fig. 2**

| | Statistical test | Factor | n | Statistic | 95% confidence interval | p value (adjusted where appropriate) | Corresponding figure |
|---|---|---|---|---|---|---|---|
| A | | | 6/group | | | | Fig. 2c |
| | Two-way mixed ANOVA | Interaction | | F (2, 20) = 4.182 | | **0.0304** | |
| | | Subregion (within subject) | | F (2, 20) = 327.2 | | **<0.0001** | |
| | | Treatment (between subjects) | | F (1, 10) = 7.933 | | **0.0183** | |
| | Sidak's multiple comparisons test | Saline vs. AMPH within Cg1 | | t(6) = 0.9744 | −7.071 to 15.94 | 0.7094 | |
| | Sidak's multiple comparisons test | Saline vs. AMPH within PrL | | t(6) = 4.036 | 6.859 to 29.87 | **0.001** | |
| | Sidak's multiple comparisons test | Saline vs. AMPH within IL | | t(6) = 1 | −6.954 to 16.05 | 0.6928 | |
| B | | | Saline = 9 AMPH = 10 | | | | Fig. 2d |
| | Two-way mixed ANOVA | Interaction | | F (13, 221) = 1.117 | | 0.3459 | |
| | | Session (within subject) | | F (13, 221) = 10.12 | | **<0.0001** | |
| | | Treatment (between subjects) | | F (1, 17) = 0.9072 | | 0.3542 | |
| C | | | Saline = 9 AMPH = 10 | | | | Fig. 2d (inset) |
| | Two-tailed unpaired t test | AUC | | t(17) = 0.326 | −3.416 to 2.502 | 0.7484 | |
| D | | | Saline = 8 AMPH = 7 | | | | Fig. 2e |
| | Two-tailed unpaired t test | Upper asymptote | | t(13) = 1.962 | −0.01218 to 0.2534 | *0.0715* | |
| E | | | Saline = 8 AMPH = 7 | | | | Fig. 2f |
| | Two-tailed unpaired t test | M50 (days) | | t(13) = 0.1289 | −2.356 to 2.091 | 0.8994 | |
| F | | | Saline = 8 AMPH = 7 | | | | Fig. 2g |
| | Two-tailed unpaired t test | Lower asymptote | | t(13) = 0.6521 | −0.2093 to 0.1122 | 0.5257 | |
| G | | | 5/group | | | | Fig. 2h |
| | Two-way mixed ANOVA | Interaction | | F (2, 16) = 1.771 | | 0.202 | |
| | | Subregion (within subject) | | F (1.183, 9.465) = 407.7 | | **<0.0001** | |
| | | Treatment (between subjects) | | F (1, 8) = 0.01415 | | 0.9082 | |
| H | | | Saline = 7 AMPH = 12 | | | | Fig. 2i |
| | Two-way mixed ANOVA | Interaction | | F (13, 221) = 1.206 | | 0.2764 | |
| | | Session (within subject) | | F (13, 221) = 18.37 | | **<0.0001** | |
| | | Treatment (between subjects) | | F (1, 17) = 0.1137 | | 0.74 | |
| I | | | Saline = 7 AMPH = 12 | | | | Fig. 2i (inset) |
| | Two-tailed unpaired t test | AUC | | t(17) = 0.8857 | −1.521 to 3.721 | 0.3882 | |
| J | | | Saline = 6 AMPH = 10 | | | | Fig. 2j |
| | Two-tailed unpaired t test | Upper asymptote | | t(14) = 0.2387 | −0.2176 to 0.174 | 0.8148 | |
| K | | | Saline = 6 AMPH = 10 | | | | Fig. 2k |
| | Two-tailed unpaired t test | M50 (days) | | t(14) = 0.1762 | −2.21 to 1.874 | 0.8627 | |
| L | | | Saline = 6 AMPH = 10 | | | | Fig. 2l |
| | Two-tailed unpaired t test | Lower asymptote | | t(14) = 0.7123 | −0.4934 to 0.2474 | 0.488 | |

Significant p values are noted in bold text.

## Netrin-1 downregulation

Female mice received bilateral microinjections of a lentivirus expressing an shRNA against Netrin-1 or a scrambled control sequence into the NAc (+1.5 AP, 2.6 ML, −3.85 DV, 30° angle, 0.5 ul per hemisphere)[62]. Mice were then treated with AMPH or saline during mid-adolescence, PND 35 ± 1 to PND 44 ± 1, and then left alone until their perfusion at PND75 ± 15.

## Data analysis

Planned comparisons were made between treatment groups for each experiment. Sex as a biological variable was included as a between-subjects factor when appropriate. Neuroanatomical data were analyzed using two-way mixed-design ANOVAs with treatment as a between-subjects factor and subregion as a within-subjects factor, with the

**Table 3 | Detailed statistics for Fig. 3**

| | Statistical test | Factor | n | Statistic | 95% confidence interval | p value (adjusted where appropriate) | Corresponding figure |
|---|---|---|---|---|---|---|---|
| A | Two-tailed unpaired t test | Netrin-1:Tubulin Fold Change | 6/group | t(10) = 3.499 | -0.8213 to -0.1823 | **0.0057** | Fig. 3e |
| B | Two-tailed unpaired t test | Netrin-1:Tubulin Fold Change | 6/group | t(10) = 0.8927 | -0.4246 to 0.1817 | 0.393 | Fig. 3f |
| C | Two-tailed unpaired t test | Netrin-1:Tubulin Fold Change | 6/group | t(10) = 0.04532 | -0.5956 to 0.5719 | 0.9647 | Fig. 3h |
| D | Two-tailed unpaired t test | Netrin-1:Tubulin Fold Change | 6/group | t(10) = 2.829 | 0.1585 to 1.333 | **0.0179** | Fig. 3i |
| E | Pearson r | | 6 | r=-0.8302, r2=0.6892 | -0.9809 to -0.05703 | **0.0408** | Fig. 3h |
| F | | | Scrambled + Saline = 5 Scrambled + AMPH = 5 Netrin-1 shRNA + Saline = 4 Netrin-1 shRNA + AMPH = 5 | | | | Fig. 3i |
| | Generalized Estimating Equations analysis (GEE) | Drug (between subjects) | | Wald Chi-Square = 2.281 (df = 1) | 0.131 | | |
| | | Virus (between subjects) | | Wald Chi-Square = 6.884(df = 1) | **0.009** | | |
| | | Region (within subjects) | | Wald Chi-Square=1546.853 (df = 2) | **<0.001** | | |
| | | Drug × Virus | | Wald Chi-Square = 0.589 (df = 1) | 0.443 | | |
| | | Virus × Region | | Wald Chi-Square = 2.083 (df = 2) | 0.353 | | |
| | | Drug × Region | | Wald Chi-Square = 5.341 (df = 2) | 0.69 | | |
| | | Drug × Virus × Region | | Wald Chi-Square = 10.119 (df = 2) | **0.006** | | |
| | Dunnett's multiple comparisons test | Scrambled:Saline vs. Scrambled:Amphetamine within Cg1 | | q(15) = 0.9696 | -28.73 to 13.19 | 0.6594 | |
| | Dunnett's multiple comparisons test | Scrambled:Saline vs. Netrin-1 shRNA:Saline within Cg1 | | q(15) = 1.641 | -36.17 to 8.279 | 0.2763 | |
| | Dunnett's multiple comparisons test | Scrambled:Saline vs. Netrin-1 shRNA:Amphetamine within Cg1 | | q(15) = 1.537 | -33.28 to 8.635 | 0.323 | |
| | Dunnett's multiple comparisons test | Scrambled:Saline vs. Scrambled:Amphetamine within PrL | | q(15) = 0.518 | -34.05 to 22.79 | 0.9182 | |
| | Dunnett's multiple comparisons test | Scrambled:Saline vs. Netrin-1 shRNA:Saline within PrL | | q(15) = 0.6796 | -37.98 to 22.31 | 0.8399 | |
| | Dunnett's multiple comparisons test | Scrambled:Saline vs. Netrin-1 shRNA:Amphetamine within PrL | | q(15) = 2.746 | -58.27 to -1.426 | **0.0388** | |
| | Dunnett's multiple comparisons test | Scrambled:Saline vs. Scrambled:Amphetamine within IL | | q(15) = 0.7311 | -9.019 to 16.02 | 0.8106 | |
| | Dunnett's multiple comparisons test | Scrambled:Saline vs. Netrin-1 shRNA:Saline within IL | | q(15) = 0.5618 | -16.13 to 10.43 | 0.8993 | |
| | Dunnett's multiple comparisons test | Scrambled:Saline vs. Netrin-1 shRNA:Amphetamine within IL | | q(15) = 2.676 | -25.33 to -0.2912 | **0.0445** | |

Significant p values are noted in bold text.

**Table 4 | Detailed statistics for Fig. 4**

| | Statistical test | Factor | n | Statistic | 95% confidence interval | p value (adjusted where appropriate) | Corresponding figure |
|---|---|---|---|---|---|---|---|
| A | | | AMPH = 4, Saline = 5 | | | | Fig. 4c |
| | Two-way mixed ANOVA | Interaction | | $F_{(2, 14)} = 5.505$ | | **0.017** | |
| | | Subregion (within subject) | | $F_{(2, 14)} = 11.84$ | | **0.001** | |
| | | Treatment (between subjects) | | $F_{(1, 7)} = 21.89$ | | **0.0023** | |
| | Sidak's multiple comparisons test | Saline vs. AMPH within Cg1 | | $t_{(21)} = 1.037$ | −24.83 to 10.65 | ns | |
| | Sidak's multiple comparisons test | Saline vs. AMPH within PrL | | $t_{(21)} = 3.998$ | −45.08 to −9.605 | **<0.01** | |
| | Sidak's multiple comparisons test | Saline vs. AMPH within IL | | $t_{(21)} = 4.915$ | −51.35 to −15.88 | **<0.001** | |
| B | | | AMPH = 4, Saline = 5 | | | | Fig. 4d |
| | Two-tailed unpaired t test | treatment (unpaired) | | $t_{(7)} = 2.45$ | −7.612 to −0.1352 | **0.0441** | |
| C | | | AMPH = 4, Saline = 5 | | | | Fig. 4e |
| | Pearson r | | | $r = -0.7634, r2 = 0.5828$ | −0.9473 to −0.2015 | **0.0167** | |
| D | | | AMPH = 4, Saline = 5 | | | | Fig. 4f |
| | Two-tailed unpaired t test | treatment (unpaired) | | $t_{(8)} = 1.115$ | −23.6 to 8.219 | 0.2974 | |
| E | | | 4 mg/kg AMPH = 6, 0.5 mg/kg AMPH = 6, Saline = 6 | | | | Fig. 4g |
| | One way ANOVA | Dose | | $F_{(2, 15)} = 17.68$ | | **0.0001** | |
| | Tukey's multiple comparisons test | Saline vs. 0.5 | | $q_{(15)} = 3.107$ | −0.1387 to 0.01158 | 0.1041 | |
| | Tukey's multiple comparisons test | Saline vs. 4.0 | | $q_{(15)} = 8.321$ | −0.2454 to −0.0951 | **<0.0001** | |
| | Tukey's multiple comparisons test | 0.5 vs. 4.0 | | $q_{(15)} = 5.214$ | −0.1818 to −0.03352 | **0.0059** | |
| F | | | AMPH = 6, Saline = 4 | | | | Fig. 4j |
| | Two-way mixed ANOVA | Interaction | | $F_{(2, 16)} = 0.2576$ | | 0.7761 | |
| | | Subregion (within subject) | | $F_{(2, 16)} = 11.41$ | | **0.0008** | |
| | | Treatment (between subjects) | | $F_{(1, 8)} = 0.004941$ | | 0.9457 | |
| G | | | AMPH = 6, Saline = 4 | | | | Fig. 4k |
| | Two-tailed unpaired t test | treatment (unpaired) | | $t_{(8)} = 0.5276$ | −2.205 to 3.513 | 0.306 | |

Significant p values are noted in bold text.

**Table 5 | Detailed statistics for Fig. 5**

| | Statistical test | Factor | n | Statistic | 95% confidence interval | p value (adjusted where appropriate) | Corresponding figure |
|---|---|---|---|---|---|---|---|
| **A** | | | | | | | |
| | One-way ANOVA | sgRNA | 4/group | $F_{(4, 15)}$ = 3.857 | | **0.0239** | Fig. 5d |
| | Dunnett's multiple comparisons test | LacZ vs. Dcc 1 | | q(15) = 3.54 | −1.649 to −0.2137 | **0.0101** | |
| | Dunnett's multiple comparisons test | LacZ vs. Dcc 2 | | q(15) = 2.68 | −1.423 to 0.01245 | 0.0547 | |
| | Dunnett's multiple comparisons test | LacZ vs. Dcc 3 | | q(15) = 1.514 | −1.116 to 0.3193 | 0.3928 | |
| | Dunnett's multiple comparisons test | LacZ vs. Dcc 4 | | q(15) = 2.842 | −1.465 to −0.03021 | **0.0401** | |
| **B** | | | | | | | Fig. 5e |
| | One-way ANOVA | sgRNA multiplexes | 4/group | $F_{(4, 15)}$ = 5.618 | | **0.0057** | |
| | Dunnett's multiple comparisons test | LacZ vs. Dcc 1, 2 | | q(15) = 0.2666 | −2.353 to 1.934 | 0.9963 | |
| | Dunnett's multiple comparisons test | LacZ vs. Dcc 1,3 | | q(15) = 3.251 | −4.699 to −0.4115 | **0.018** | |
| | Dunnett's multiple comparisons test | LacZ vs. Dcc 1,4 | | q(15) = 1.104 | −3.012 to 1.276 | 0.6472 | |
| | Dunnett's multiple comparisons test | LacZ vs. Dcc 1,2,3,4 | | q(15) = 3.584 | −4.96 to −0.6729 | **0.0093** | |
| **C** | | | LacZ = 4, Dcc = 5 | | | | Fig. 5i |
| | One-tailed Unpaired t test | sgRNA multiplexes | | t(7) = 2.140 | −0.01661 to 0.3338 | **0.0348** | |
| **D** | | | LacZ = 3, Dcc = 3 | | | | Fig. 5j |
| | Two-tailed Unpaired t test | sgRNA multiplexes | | t(4) = 5.028 | 0.1294 to 0.4486 | **0.0073** | |
| **E** | | | LacZ = 3, Dcc = 3 | | | | Fig. 5k |
| | Pearson r | | | r = 0.9251, r2 = 0.8558 | 0.4554 to 0.9919 | **0.0082** | |
| **F** | | | LacZ + Saline = 9, LacZ + AMPH = 5, Dcc + Saline = 6, Dcc + AMPH = 7 | | | | Fig. 5m |
| | Generalized Estimating Equations analysis (GEE) | Drug (between subjects) | | Wald Chi-Square = 4.3 (df = 1) | | **<0.001** | |
| | | Construct (between subjects) | | Wald Chi-Square = 0.322 (df = 1) | | 0.38 | |
| | | Session (within subjects) | | Wald Chi-Square = 158.053 (df = 13) | | **<0.001** | |
| | | Drug × Construct | | Wald Chi-Square = 5.577 (df = 1) | | **0.018** | |
| | | Construct × Session | | Wald Chi-Square = 25.497 (df = 13) | | **0.02** | |
| | | Drug × Session | | Wald Chi-Square = 37.392 (df = 13) | | **<0.001** | |
| | | Drug × Construct × Session | | Wald Chi-Square = 35.645 (df = 13) | | **<0.001** | |
| **G** | | | LacZ + Saline = 9, LacZ + AMPH = 5, Dcc + Saline = 6, Dcc + AMPH = 7 | | | | Fig. 5m (inset) |
| | Two-way ANOVA | Interaction | | $F_{(1, 25)}$ = 8.37 | | **0.0078** | |
| | | sgRNA construct (between subject) | | $F_{(1, 25)}$ = 0.3124 | | 0.5812 | |
| | | Drug (between subject) | | $F_{(1, 25)}$ = 4.317 | | **0.0482** | |

**Table 5 (continued) | Detailed statistics for Fig. 5**

| Statistical test | Factor | n | Statistic | 95% confidence interval | p value (adjusted where appropriate) | Corresponding figure |
|---|---|---|---|---|---|---|
| Tukey's multiple comparisons test | LacZ:Sal vs. LacZ: Amph | | q(25) = 4.836 | -6.18 to -0.6699 | **0.0109** | |
| Tukey's multiple comparisons test | LacZ:Sal vs. Dcc:Sal | | q(25) = 3.554 | -4.982 to 0.2248 | 0.0822 | |
| Tukey's multiple comparisons test | LacZ:Sal vs. Dcc: Amph | | q(25) = 3.035 | -4.145 to 0.5118 | 0.1664 | |
| Tukey's multiple comparisons test | LacZ: Amph vs. Dcc:Sal | | q(25) = 1.361 | -1.945 to 4.037 | 0.7717 | |
| Tukey's multiple comparisons test | LacZ: Amph vs. Dcc: Amph | | q(25) = 2.271 | -1.147 to 4.363 | 0.3937 | |
| Tukey's multiple comparisons test | Dcc:Sal vs. Dcc: Amph | | q(25) = 0.8395 | -2.041 to 3.165 | 0.9331 | |
| H | | LacZ + Saline = 9; LacZ + AMPH = 5; Dcc + Saline = 7; Dcc + AMPH = 10 | | | | Fig. 5o |
| Two-way ANOVA | Interaction | | F (1, 27) = 1.069 | | 0.3103 | |
| | sgRNA construct | | F (1, 27) = 0.8237 | | 0.3721 | |
| | Drug | | F (1, 27) = 0.043 | | 0.8373 | |
| I | | LacZ + Saline = 9; LacZ + AMPH = 5; Dcc + Saline = 7; Dcc + AMPH = 10 | | | | Fig. 5p |
| Two-way ANOVA | Interaction | | F (1, 27) = 17.94 | | **0.0002** | |
| | sgRNA construct | | F (1, 27) = 0.2923 | | 0.5932 | |
| | Drug | | F (1, 27) = 7.713 | | **0.0098** | |
| Sidak's multiple comparisons test | LacZ:Sal vs. LacZ: Amph | | q(27) = 0.0004 | -9.499 to -2.326 | **0.0004** | |
| Sidak's multiple comparisons test | LacZ:Sal vs. Dcc:Sal | | q(27) = 0.0509 | -6.355 to 0.1249 | 0.0645 | |
| Sidak's multiple comparisons test | LacZ:Sal vs. Dcc: Amph | | q(27) = 0.29 | -4.84 to 1.068 | 0.3984 | |
| Sidak's multiple comparisons test | LacZ: Amph vs. Dcc:Sal | | q(27) = 0.1759 | -0.9674 to 6.562 | 0.2384 | |
| Sidak's multiple comparisons test | LacZ: Amph vs. Dcc: Amph | | q(27) = 0.0155 | 0.5053 to 7.548 | **0.0186** | |
| Sidak's multiple comparisons test | Dcc:Sal vs. Dcc: Amph | | q(27) = 0.6918 | -1.939 to 4.398 | 0.8612 | |
| J | | LacZ + Saline = 9; LacZ + AMPH = 5; Dcc + Saline = 7; Dcc + AMPH = 10 | | | | Fig. 5q |
| Two-way ANOVA | Interaction | | F (1, 25) = 2.973 | | 0.097 | |
| | sgRNA construct | | F (1, 25) = 3.006 | | 0.0953 | |
| | Drug | | F (1, 25) = 18.44 | | **0.0002** | |

Significant p values are noted in bold text.

exception of the Netrin-1 shRNA experiment, which used a Generalized Estimating Equations (GEE) approach to account for 2 independent (Virus and Treatment) and 1 repeated (Subregion) factors. To quantify the complexity of PFC neuron dendritic arbors, we used the Dendritic Complexity Index (DCI)[117]. Behavioral data from the CPP test was analyzed using Student's *t* tests or one-way ANOVA. Data from the Go/No-Go task across sessions was analyzed using GEE while area under the curve was assessed using Student's *t* tests or two-way ANOVA. Sigmoidal curve fitting was performed in MATLAB, by fitting commission errors to session with a sigmoid of the form $y = Min + (Max - Min)/1 + 10^{(x50 - x)}*p$ where Min is the lower asymptote, Max is the upper asymptote, x50 is the position parameter denoting the training day at which the slope of the curve is maximal, and p determines the steepness of the sigmoid curve. The resulting fit was used to derive an index of improvement in the commission errors, defined as the day sustaining a half-maximal rate of commission errors (M50). For qPCR and Western blot experiments, Student's *t* tests or one-way ANOVA were used to assess treatment effect. Correlations were evaluated using Pearson's r and linear regression. When representations of data were normalized for graphs, all statistics were performed on the raw data. When post hoc testing was used, the most appropriate correction for multiple comparisons was chosen based on the factor design of the ANOVA in order to maximize power and not violate statistical assumptions. A Tukey's multiple comparisons test was used when all samples were independent and all possible interactions were considered. Sidak's multiple comparisons test was used when all samples were independent but comparisons were made only within one factor of interest, as comparing all groups was redundant or irrelevant. Dunnett's multiple comparisons test was used when conditions were compared to an explicit control condition, such as the *LacZ* sgRNA construct which should not amplify endogenous genes in the CRISPRa experiments. Detailed information about all statistical tests used are presented in Tables 1–5 for the main figures, and Supplementary Tables 2–5 for Supplementary data figures. All statistical analyses were carried out using Prism software (GraphPad), with the exception of the GEE, which was done in SPSS.

### Reporting summary
Further information on research design is available in the Nature Portfolio Reporting Summary linked to this article.

## Data availability
Uncropped blot images are available in Supplementary Data Fig. 3. Source data are provided with this paper.

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

## Acknowledgements

The authors dedicate this work to the late Dominique Nouel – an excellent and dedicated scientist, a kind mentor and colleague, and a generous and loyal friend. Dr. Helen Cooper of the University of Queensland, Brisbane, QLD, Australia provided the polyclonal rabbit anti-DCC antibody. This work was supported by the National Institute on Drug Abuse at the National Institutes of Health (R01DA037911 to C.F.; F31DA041188 to L.M.R.), the Natural Sciences and Engineering Research Council of Canada (RGPIN-2020-04703 to C.F.), the Canadian Institutes of Health Research (FRN: 156272 to C.F.), L.M.R. was also supported by predoctoral fellowships from The Djavad Mowafaghian Foundation and Fulbright Canada, and a post-doctoral NIDA-Inserm Drug Abuse Research Fellowship. J.M.R.L was supported by a Doctoral Training fellowship from the Fonds de Recherche du Québec en Santé and a Graduate Student Fellowship from the Healthy Brains for Healthy Lives initiative of the Canada First Research Excellence Fund at McGill University, Canada. The authors declare no conflicts of interest.

## Author contributions

L.M.R. and C.F. designed the studies. L.M.R., C.P., J.M.R.L., S.I., T.O., and J.G.E. performed behavioral experiments. G.H., D.N., S.C., J.M.R-L, M.G., and R.G.A. performed qPCR and western blotting experiments. L.M.R., D.M., D.N., J.Z., M-Y. H., Z.N., and A.H.P.U. performed immunohistochemistry and stereological experiments. M.P., M.W. and B.K. performed Golgi experiments. G.H. and S.B.N. performed CRISPR validation studies, with in vitro work done in the lab of L-É. T. J.J.D. provided CRISPRa virus material and the sgRNAs, based on sgRNAs designed by K.S. and L.M.R. G.H. performed CRISPR virus injections and behavioral experiments. L.M.R, G.H., C.P., and C.F. analyzed the data. L.M.R. and C.F. wrote the manuscript. All authors discussed the results, edited, and approved the manuscript.

## Competing interests

The authors declare no competing interests.
