## [Peer Review File · Nature Communications]

Amphetamine disrupts dopamine axon growth in adolescence by a sex-specific mechanism in miceREVIEWER COMMENTS

Reviewer #1 (Remarks to the Author):

Reynolds et al. present an interesting paper, “Amphetamine disrupts dopamine axon growth in adolescence by a sex-specific mechanism”, indicating that DCC/Netrin is important for regulating the age- and sex-specific effects of amphetamine on dopaminergic neurodevelopment and related behavior. This report is an important step forward in understanding possible mechanisms underlying sex-specific adolescent vulnerabilities to addictive substances. In short, early adolescent amphetamine exposure reduced VTA DCC and NAc Netrin in males, resulting in long-term PFC dopaminergic and PFC-related behavioral changes in adulthood. Conversely, mid-adolescent amphetamine exposure reduced VTA DCC but increase NAc Netrin in females, with no apparent long-term consequences. There are many strengths, including well-written prose, a well-validated behavioral task, two different adolescent ages, two different amphetamine doses, the inclusion of both sexes, and a new Crispr-mediated DCC overexpression tool to examine causality. There are, however, some holes that need to be addressed before this reviewer can endorse publication in Nat. Comms. In particular, the female data is under-developed. This is framed as a sex differences paper (as it should be!) and much of the novelty therein derives from the highly sexually divergent findings. But many of the later studies are only performed in males, testing important, but relatively ‘safe’ hypotheses based on the authors previous body of work. If the female data is better developed, this is a lovely and impactful paper.

Major critiques:

1. The authors implicate increased NAc netrin levels in female provide ‘protection’ against mid-adolescent amphetamine exposure, but there is no evidence (unlike the data in males) that DCC/Netrin would be playing a prominent role in female dopaminergic neurodevelopment. To make this claim, one would need to reduce Netrin in the NAc and show that is causally implicated in long-term behavioral abnormalities. That would then also recapitulate the male-like DCC/Netrin vulnerability pattern, which would be a nice tie-in to discuss sexual dimorphism more concretely.
2. The Netrin data is reported to come from the same animals as the DCC data – are those values correlated within animal? This would be additional evidence for DCC/Netrin working as a system.
3. I appreciate the full Western blots in the Supplemental, but there looks to be quite a large quality difference between the early and mid- adolescent blots. Additionally, there looks like there is a noticeable shift in the tubulin molecular weight in the mid-adolescent blots. How is this explained? Please also verify that tubulin is an appropriate control and does not change across treatment groups.
4. Therapeutic amphetamine dose experiments should also be completed in mid-adolescent females, as DCC/Netrin is also shown to be regulated by recreational amphetamine, which was the impetus for those studies in males.
5. In Fig. 4k, the NAc varicosities in the saline group looks very different from the varicosities in the saline group in Fig. 4d. How is this explained? Which is more representative? Are tracer efficacies different between the two experiments?
6. The DCC overexpression studies are compelling, but are not mimicking “female-like protection” (Abstract). The proposed female protection according to the authors is from Netrin overexpression in

the NAc (though this is not proven, as pointed out in #1). It is simply 'protective' and should not be compared to the female mechanism. If the authors wish to upregulate Netrin in males to mimic the female pattern, and that confers protection, then this statement would stand. While this reviewer thinks that symmetry with a female vulnerability experiment (#1) would provide the most compelling statement for sexual dimorphism, I will not ask for those studies because a set of rescue studies were performed in males.

Minor critiques:

7. Please indicate in the histogram that there is a sex x interaction treatment in Fig. 1j.
8. End of second Results paragraph "These results identify..." is an unnecessary sentence and unlikely to be factually correct. The significance of the paper does not hinge on that sentence.
9. Could you please add more explanation of the intersectional viral tracing technique in Fig. 4 to the main text? I read it in the supplemental methods, but more detail would be nice to better understand the figure.

Reviewer #2 (Remarks to the Author):

General Thoughts:

The manuscript by Reynolds et al., investigates how disruption of DCC/netrin signaling during adolescence impacts dopaminergic axonal growth and behavior. The authors found that amphetamine during adolescence disrupts axonal outgrowth of DA neurons in males and their performance on a go-no/go task. Females on the other hand are protected against the effects of adolescent amphetamine exposure even though they show changes in Dcc and mir218 after mid-adolescent exposure to amphetamine. The authors use Crisp dCas9 technology to upregulate Dcc in the VTA of males during adolescence and show similar effects on DA axonal innervation of the PFC and behavior. These data are exciting and would be impactful but sex-specific effects combined with the male specific endpoints make the manuscript a little disjointed and make it a little difficult to identify a clear take-away.

Comments:

1) The most noteworthy results presented are 1) that females are protected from the effects of adolescent amphetamine exposure, potentially through the concomitant upregulation of netrin protein in the NAC and 2) that upregulation of DCC through the use of Crispr/dCas9 technologies recapitulates the effects of adolescent amphetamine in males.

While both findings are of interest to the field, the female effects are the most novel given that the authors have already published the male specific effects in the past. This study uses new tools to further link DCC to the changes in dopaminergic innervation and behavior. However, there are specific experiments that would make the manuscript very impactful. First, a more thorough investigation of the

female specific effects. There are several unanswered questions that need to be addressed. For example, are the same netrin and dcc levels observed in females and males. The authors mention that “all dopamine axons in the NAC express DCC” but this work was done in males and given the effects it shouldn’t be assumed that the same expression levels will be observed in females. Without these data the findings are incomplete.

2) It seems that the authors are presenting 2 very different stories and therefore need to provide evidence for 2 different sets of conclusions. The male-specific work (Figures 4 & 5) are supported and much of the data presented here replicates work that has been previously published by this group. The female effects need more experiments to fully support their conclusions. The female specific effects described are a rather cursory evaluation of the sex-specific effects of adolescent amph exposure. It is incredibly difficult to tie the two findings together into a complete picture. The authors could use the sex-specific effects to advance our knowledge of how the DCC system impacts development of the mesocorticolimbic system across the sexes. For example, the authors could upregulate netrins in the NAC in males and females to see if they replicate the female specific effects and if those effects are protective against early adolescent AMPH exposure in males.

3) The data analysis appears correct but there are some flaws in the interpretation and conclusions. The biggest flaw is in the assumption that DCC and netrin in dopamine neurons is the same across the sexes. The studies investigating DCC in dopamine neurons were done in males (reference #s 61 & 62). The authors show here that DCC/netrin effects are sex-specific but assume that the expression patterns will be the same. This is not necessarily the case. It would be helpful for the authors to do a full assessment of DCC and netrin in females before they make claims about the mechanisms regulating protection from AMPH exposure in females. The conclusions and interpretation should at least be revised to reflect that lack of studies in females and that many of the interpretations are based on findings that have only been done in males.

An example of this assumption is provided here:

“In the PFC, mNetrin-1 levels are high but only few dopamine axons express DCC. In contrast, Netrin-1 levels are lower in the nucleus accumbens (NAc), but all dopamine axons in this region highly express DCC (Fig 3a).⁶⁶ Furthermore, dopamine axons are the only source of DCC protein expression in the NAc,⁶⁶ suggesting a crucial and complementary role of DCC in dopamine axons and Netrin-1 in the NAc in the developmental organization of mesocorticolimbic dopamine connectivity. Finally, Netrin-1 levels in the NAc decline across adolescence in male mice,⁶¹”

These two studies were only done in males. It is unclear that the authors can assume these would be the same in females, especially given the sex-specific effects already observed.

4) Generally speaking, the methodology is seemingly sound. However, there are a few questions that need to be addressed to help assess this. First, in Figure 4g, can the authors please explain what the “delta place preference” means? One would assume the author showing the difference between a

pretest “preference” and a posttest preference. However, the units don’t necessarily make sense with that calculation. Is this a differences in percent time?

5) Additionally, in figure 4, can the authors please explain why there appears to be such big differences in their saline animals between figure 4b and 4i?

6) On the stats table, the last figure is mislabeled – it should be figure 5 but is labeled as figure 2.

Reviewer #3 (Remarks to the Author):

This is an excellent manuscript describing a dcc-netrin mechanism by which early adolescent amphetamine (AMPH) exposure causes VTA dopaminergic neurons to ectopically target the PFC instead of the NAcc in males, with consequential deficits in adult impulse responsivity. Interestingly, females show reduced dcc in the VTA following late adolescent, rather than early adolescent AMPH exposure, but this did not translate to impulsivity in adulthood. This was attributed to increased nestin expression in the NAcc, which compensated for decreased dcc and allowed proper targeting of DA neurons to the NAcc. Methodology used was innovative and robust, and data was analyzed and interpreted appropriately. The methods in the expanded section were sufficiently detailed (I assume that inclusion in the main manuscript is not required, though this is an editorial question). The results are highly significant, as they are the first to demonstrate, using a molecular- region-, and time-specific prevention of a behavior that is strongly predictive of drug addiction following adolescent drug exposure, and clearly demonstrates a mechanism underpinning male vulnerability to drug addiction following adolescent exposure as well. I did not find much to substantively critique about these studies or the report. There were, however, several unfortunate groups/experiments lacking which would have completed the story of sex-specificity, and a few other minor issues that if addressed could improve the overall manuscript.

1. Figure 1i: The graph illustrates and results section reports that miR-218 is negatively correlated with Dcc mRNA in females, but the figure caption states that they were “positively associated”

2. Figure 2 and results: While it is appreciated that previous work by this group (and figure 5 of this paper) showed male impulsivity following adolescent amphetamine, reporting a sex-specific effect of adolescent amphetamine on enduring behavior requires a simultaneous demonstration of male/female data in the same experiment with the same strain of mice, and for sex to be analyzed as a biological variable (as done in the other experiments reported in the manuscript).

3. Figure 4: It is not clear to me that ventral PFC regions were observed to be more innervated by TH+ terminals in the extended data (there does not seem to be a statistical subregion effect), as suggested in the results (when it is stated that the YPF+ results were consistent with overall DA innervation).

4. Similar to point 2, it is unfortunate that female DA innervation of NAcc and PFC were not reported; although the behavioral and dcc/nestin data suggest that innervation is not altered in females, this remains a question unless it is measured.

5. The data (YFP+ innervation of PFC, and negative correlation between NAc and PFC terminals) is fascinating. Do the authors believe that these axons had once reached the NAc in early adolescence (in order to be retrogradely labeled) and then rerouted to the PFC during adolescence? Has this ever been seen before, i.e., rather than a mis-direction, there is actually a re-direction once a target is already reached? A discussion would be interesting about the potential time course of possibilities for intervention while the axons are moving to a new destination.

6. Figure 5: It appears that the lack of difference between Sal-treated and AMPH-treated dcc overexpressing mice was due to a “meeting in the middle” between lacZ sgRNA SAL- and AMPH-treated animals, whereby SAL-treated dcc-overexpressing mice showed higher impulsivity. The authors should address this.

Minor:

The first sentence, “adolescence is a conserved period of life”, is unclear in its meaning—do the authors mean to say it is evolutionarily conserved?

In the following section, we address each of the comments raised by the reviewers. We highlight changes to the text in red throughout the manuscript file.

REVIEWER COMMENTS

Reviewer #1 (Remarks to the Author):

Reynolds et al. present an interesting paper, “Amphetamine disrupts dopamine axon growth in adolescence by a sex-specific mechanism”, indicating that DCC/Netrin is important for regulating the age- and sex-specific effects of amphetamine on dopaminergic neurodevelopment and related behavior. This report is an important step forward in understanding possible mechanisms underlying sex-specific adolescent vulnerabilities to addictive substances. In short, early adolescent amphetamine exposure reduced VTA DCC and NAc Netrin in males, resulting in long-term PFC dopaminergic and PFC-related behavioral changes in adulthood. Conversely, mid-adolescent amphetamine exposure reduced VTA DCC but increase NAc Netrin in females, with no apparent long-term consequences. There are many strengths, including well-written prose, a well-validated behavioral task, two different adolescent ages, two different amphetamine doses, the inclusion of both sexes, and a new Crispr-mediated DCC overexpression tool to examine causality. There are, however, some holes that need to be addressed before this reviewer can endorse publication in Nat. Comms. In particular, the female data is under-developed. This is framed as a sex differences paper (as it should be!) and much of the novelty therein derives from the highly sexually divergent findings. But many of the later studies are only performed in males, testing important, but relatively ‘safe’ hypotheses based on the authors previous body of work. If the female data is better developed, this is a lovely and impactful paper.

Major critiques:

1. The authors implicate increased NAc netrin levels in female provide ‘protection’ against mid-adolescent amphetamine exposure, but there is no evidence (unlike the data in males) that DCC/Netrin would be playing a prominent role in female dopaminergic neurodevelopment. To make this claim, one would need to reduce Netrin in the NAc and show that is causally implicated in long-term behavioral abnormalities. That would then also recapitulate the male-like DCC/Netrin vulnerability pattern, which would be a nice tie-in to discuss sexual dimorphism more concretely.

We thank the reviewer for bringing up this important point. We have now added new data and made several changes to the text in order to clarify why we believe that DCC in dopamine neurons plays an important role in both male and female dopamine development.

1) We now explicitly state in the text that *DCC* is enriched in VTA dopamine neurons of both male and female rodents and that the percentage of dopamine neurons expressing *Dcc* does not differ between the sexes. This is evidence published recently from the lab of our collaborator Jeremy Day (Phillips et al., *Cell Reports* 2022). We have also added male and female symbols to the diagram in Figure 1a to clarify that this is the case for both sexes.

i) Figure 1a

ii) Figure 1 Legend, page 12:

(a) *Dcc* mRNA is expressed by 99% of dopamine neurons in the VTA of both male and female mice.⁵⁹

iii) Results section, page 3:

“The guidance cue receptor DCC is highly enriched in dopamine neurons of the ventral tegmental area in both male and female rodents, with no difference between sexes in the percentage of dopamine neurons expressing Dcc mRNA (VTA; Fig 1a,b).”

2) We have now added new neuroanatomical data (Extended data Figure 1) showing that adult female mice have the exact same segregation pattern of DCC protein expression between mesolimbic and mesocortical dopamine axons we have previously reported in adult male mice (Manitt et al., *J Neurosci* 2012). The fact that nucleus accumbens, but not prefrontal cortex, dopamine axons express DCC suggests that the overall role of DCC in segregating mesolimbic and mesocortical pathways is the same regardless of sex. However, we want to emphasize that the timing of this event may differ between the sexes, given that adolescence, and puberty in particular, are not perfectly aligned across chronological age in males and females.

Ongoing studies from our lab are currently focused on fully characterizing male and female dopamine developmental timelines. These studies include detailed assessment of DCC and Netrin-1 expression patterns across normative development. At this point, it would be premature to include the results we have gathered so far in this regard, because we cannot make definitive conclusions yet. In addition, the focus of this paper is the age- and sex-specific regulation of DCC and Netrin-1 in dopamine systems in response to the same adolescent drug experience and these other data will be outside of this main theme.

i) Extended data Figure 1

ii) Extended data Figure 1 Legend, Supplemental material page 9:

Extended data figure 1 *DCC protein is highly expressed in mesolimbic, but not mesocortical dopamine axons of adult female mice.* (a) DCC protein is highly expressed in the NAc of adult female mice, with its expression limited to dopamine axons innervating this region. (b) In the PFC, DCC receptors are expressed by local neurons,¹⁶ but they are rarely or not expressed by dopamine axons. This segregation of DCC expression between the mesolimbic and mesocortical dopamine pathways is identical to what has been previously reported in male mice.¹⁰

iii) We have also added the following information to the Results section, page 3:

“All DCC protein expression in the NAc of female mice is localized to dopamine axons (Extended data Fig 1a). In contrast, few or no dopamine axons in the PFC express DCC receptors (Extended data Fig 1b). The exact same segregation pattern of DCC expression is observed in male mice.⁶⁰”

3) Regarding Netrin-1 upregulation in mid-adolescent females, we have now added additional data to Figure 3 indicating that AMPH-induced upregulation of Netrin-1 in the NAc of mid-adolescent female mice has a protective effect:

i) Figure 3h – We conducted correlational analysis to assess the relationship between *Dcc* mRNA expression in the VTA and Netrin-1 protein levels in the NAc in all of our treatment groups. This analysis revealed a strong and significant *negative* correlation only in the female mice treated with AMPH in mid-adolescence. This result suggests that in mid-adolescent females, drug-induced *downregulation* of *Dcc* mRNA in dopamine axons is tightly associated with the drug-induced *upregulation* of Netrin-1 in the NAc. These results are now included in panel h of Figure 3.

ii) Figure 3 Legend, page 16:

*“(h) *Dcc* mRNA expression in the VTA and Netrin-1 protein levels in the NAc of female mice treated with AMPH in mid-adolescence show a strong and significant negative correlation, suggesting that drug-induced downregulation of *Dcc* mRNA in dopamine axons is tightly associated with the drug-induced upregulation of Netrin-1 in the NAc (Table 3E). ”*

iii) Detailed statistics Table 3E :

	Statistical test	Factor	n	Statistic	95% confidence interval	p value (adjusted where appropriate)	corresponding figure
E			6				Figure 3h
	Pearson r			$r = -0.8302,$ $r^2 = 0.6892$	-0.9809 to - 0.05703	0.0408	

iv) There is no correlation between VTA *Dcc* mRNA levels and NAc Netrin-1 expression in all the other groups. These results are included in Extended Figure 3.

v) Extended data 3 Figure Legend, Supplemental data page 11 :

“(c) There was no correlation between Netrin-1 protein in the NAc and Dcc mRNA in the VTA for mice treated with AMPH or Saline in early adolescence, regardless of sex (Extended data Table 3 A). (d) There was no correlation between Netrin-1 protein in the NAc and Dcc mRNA in the VTA for mid-adolescent males or female mice treated with saline, nor in mid-adolescent males treated with AMPH (Extended data Table 3 B).”

vi) Extended data Table 3:

	Statistical test	Factor	n	Statistic	95% confidence interval	p value (adjusted where appropriate)	corresponding figure
A							Extended data Figure 3c
	Pearson r	Males + saline	4	r = 0.5482, r2 = 0.3005	-0.8727 to 0.9885	0.4518	
		Females + saline	4	r = -0.1112, r2 = 0.01236	-0.9688 to 0.9516	0.8888	

		Males + AMPH	5	r = -0.04895, r2 = 0.002396	-0.8927 to 0.8709	0.9377	
		Females + AMPH	6	r = 0.5152, r2 = 0.2654	-0.5094 to 0.9356	0.2956	
B							Extended data Figure 3d
	Pearson r	Males + saline	7	r = -0.02616, r2 = 0.0006842	-0.7642 to 0.7415	0.9556	
		Females + saline	6	r = 0.154, r2 = 0.02373	-0.7515 to 0.8583	0.7708	
		Males + AMPH	6	r = 0.2672, r2 = 0.07138	-0.6951 to 0.8865	0.6088	

vii) Figure 3i – To follow up these correlational results, we now add results from a new functional neuroanatomical experiment showing that the upregulation of Netrin-1 in the NAc of female mice treated with AMPH in mid-adolescence indeed protects against the effects of AMPH on dopamine connectivity. To accomplish this, we downregulated *Netrin-1* in the NAc of female mice using shRNA before the AMPH or saline treatment in mid-adolescence. We found that only female mice with *Netrin-1* shRNA + AMPH treatment had an increased expanse of the dopamine input to the PFC in adulthood.

Increased dopamine input volume in the PFC in adulthood is a proxy of ectopic growth of mesolimbic dopamine axons in the PFC. In fact, this finding strongly resembles the effect we have seen in male mice treated with AMPH in early adolescence, which is shown in the current manuscript (Extended data Figure 4a) as well as in our previous work (Reynolds et al., *Neuropsychopharmacology* 2014). These new data support the idea that AMPH-induced upregulation of Netrin-1 in the NAc of mid-adolescent females is protective.

F			Scrambled + Saline= 5 Scrambled + AMPH= 5 Netrin-1 shRNA + Saline = 4 Netrin-1 shRNA + AMPH = 5				Figure 3i
	Generalized Estimating Equations analysis (GEE)	Drug (between subjects)		Wald Chi-Square = 2.281 (df = 1)		0.131	
		Virus (between subjects)		Wald Chi-Square = 6.884(df = 1)		0.009	
		Region (within subjects)		Wald Chi-Square = 1546.853 (df = 2)		<0.001	
		Drug x Virus		Wald Chi-Square = 0.589 (df = 1)		0.443	
		Virus x Region		Wald Chi-Square = 2.083 (df = 2)		0.353	
		Drug x Region		Wald Chi-Square = 5.341 (df = 2)		0.69	
		Drug x Virus x Region		Wald Chi-Square = 10.119 (df = 2)		0.006	
	Dunnett's multiple comparisons test	Scrambled:Saline vs. Scrambled:Amphetamine within Cg1		q(15) = 0.9696	-28.73 to 13.19	0.6594	
	Dunnett's multiple comparisons test	Scrambled:Saline vs. Netrin-1 shRNA:Saline within Cg1		q(15) = 1.641	-36.17 to 8.279	0.2763	
	Dunnett's multiple comparisons test	Scrambled:Saline vs. Netrin-1 shRNA:Amphetamine within Cg1		q(15) = 1.537	-33.28 to 8.635	0.323	
	Dunnett's multiple comparisons test	Scrambled:Saline vs. Scrambled:Amphetamine within PrL		q(15) = 0.518	-34.05 to 22.79	0.9182	

	Dunnett's multiple comparisons test	Scrambled:Saline vs. Netrin-1 shRNA:Saline within PrL	q(15) = 0.6796	-37.98 to 22.31	0.8399	
	Dunnett's multiple comparisons test	Scrambled:Saline vs. Netrin-1 shRNA:Amphetamine within PrL	q(15) = 2.746	-58.27 to -1.426	0.0388	
	Dunnett's multiple comparisons test	Scrambled :Saline vs. Scrambled :Amphetamine within IL	q(15) = 0.7311	-9.019 to 16.02	0.8106	
	Dunnett's multiple comparisons test	Scrambled :Saline vs. Netrin-1 shRNA:Saline within IL	q(15) = 0.5618	-16.13 to 10.43	0.8993	
	Dunnett's multiple comparisons test	Scrambled :Saline vs. Netrin-1 shRNA:Amphetamine within IL	q(15) = 2.676	-25.33 to -0.2912	0.0445	

x) We have now made the following modifications to the Results section, page 5:

“In addition, we find a strong, negative correlation between Dcc mRNA in the VTA and Netrin-1 protein in the NAc of females treated with AMPH during mid-adolescence: the greater the downregulation of Dcc levels in the VTA by AMPH, the greater the drug-induced upregulation of Netrin-1 in the NAc (Fig 3h). None of the other groups studied showed a correlation between Dcc in the VTA and Netrin-1 in the NAc (Extended data Fig 3). Therefore, upregulation of Netrin-1 in females may be a compensatory effect of drug treatment, protecting against enduring consequences triggered by drug-induced Dcc downregulation.

To test this idea, we used an shRNA approach to downregulate Netrin-1 expression in the NAc of female mice before subjecting them to AMPH or saline treatment in mid-adolescence (Figure 3i). In adulthood we stereologically quantified the expanse of the dopamine innervation to their PFC. In mid-adolescence, AMPH had no effect on the expanse of the dopamine input to the PFC in mice that received bilateral microinfusion of scrambled shRNA, in agreement with our previous results (Figure 2h). In contrast, AMPH exposure in mid-adolescence produced an increase in the volume of the PFC dopamine input in when it was paired with Netrin-1 downregulation in the NAc (Figure 3i). This increase in PFC dopamine innervation volume mimics our previous results in male mice treated in early adolescence,⁶⁷ providing the first evidence that Netrin-1 upregulation in the NAc of mid-adolescent female mice compensates for the downregulation of Dcc in the VTA and protects females against the deleterious effects of AMPH on adolescent mesocorticolimbic dopamine development.”

2. The Netrin data is reported to come from the same animals as the DCC data – are those values

E			6				Figure 3h
	Pearson r			$r = -0.8302$, $r^2 = 0.6892$	-0.9809 to -0.05703	0.0408	

iv) The VTA *Dcc* mRNA and NAc Netrin-1 correlation was negative for all the other groups. These results are included in Extended Figure 3.

v) Extended data 3 Figure Legend, Supplemental data page 11 :

“(c) There was no correlation between Netrin-1 protein in the NAc and Dcc mRNA in the VTA for mice treated with AMPH or Saline in early adolescence, regardless of sex (Extended data Table 3 A). (d) There was no correlation between Netrin-1 protein in the NAc and Dcc mRNA in the VTA for mid-adolescent males or female mice treated with saline, nor in mid-adolescent males treated with AMPH (Extended data Table 3 B).”

vi) Extended data Table 3:

	Statistical test	Factor	n	Statistic	95% confidence interval	p value (adjusted where appropriate)	corresponding figure
A							Extended data Figure 3c
	Pearson r	Males + saline	4	r = 0.5482, r2 = 0.3005	-0.8727 to 0.9885	0.4518	
		Females + saline	4	r = -0.1112, r2 = 0.01236	-0.9688 to 0.9516	0.8888	
		Males + AMPH	5	r = -0.04895, r2 = 0.002396	-0.8927 to 0.8709	0.9377	
		Females + AMPH	6	r = 0.5152, r2 = 0.2654	-0.5094 to 0.9356	0.2956	
B							Extended data Figure 3d
	Pearson r	Males + saline	7	r = -0.02616, r2 = 0.0006842	-0.7642 to 0.7415	0.9556	
		Females + saline	6	r = 0.154, r2 = 0.02373	-0.7515 to 0.8583	0.7708	
		Males + AMPH	6	r = 0.2672, r2 = 0.07138	-0.6951 to 0.8865	0.6088	

3. I appreciate the full Western blots in the Supplemental, but there looks to be quite a large quality difference between the early and mid- adolescent blots. Additionally, there looks like there is a noticeable shift in the tubulin molecular weight in the mid-adolescent blots. How is this explained? Please also verify that tubulin is an appropriate control and does not change across treatment groups.

We agree that there are qualitative differences between the two western blots, which most likely results from the fact that they were performed by two different researchers in the lab, who may have slight differences in the specificities of their technique. These small differences in technique can be seen (i) from top left corner snips of the membrane of P21 mice, but not of p35 mice; (ii) in the differences in the ordering of the samples. Importantly, all of the major parts of the protocol, including incubation times and antibody concentrations, were the same across the two experiments. To minimize any impact of potential qualitative differences between experimenters, we make comparisons only within, and never between, the two blots.

Regarding the shift in the alpha-tubulin molecular weight in P35, we notice that all bands starting with lane 2 are shifted down. We attribute this to an imperfection in gel polymerization during gel casting. Alpha-tubulin, our chosen loading control, is a high expression housekeeping gene which is not known to be affected by amphetamine treatment. This is in contrast to actin – a popular loading control which we have previously seen to be altered following amphetamine treatment. We consider alpha-tubulin an appropriate control for this experiment, and we habitually use this as a control in

our western blot experiments (see Reynolds et al 2018, Cuesta et al 2019, Cuesta et al 2020).

To highlight the consistency in our Netrin-1 blot outcomes, we are including in this letter an additional, recent Netrin-1 blot from our group where we assessed multiple experimental conditions in NAc samples derived from mid-adolescent female mice. The lanes with the same treatment groups as in the experiments presented in this manuscript (namely, exposure to saline or to 4 mg/kg amphetamine in mid-adolescent female

mice) are highlighted within dashed boxes.

4. Therapeutic amphetamine dose experiments should also be completed in mid-adolescent females, as DCC/Netrin is also shown to be regulated by recreational amphetamine, which was the impetus for those studies in males.

Indeed, the impetus for assessing the Adderall-like dose of 0.5 mg/kg amphetamine (ALD) was because our previous work has shown that it does not downregulate *Dcc* expression in the VTA of early adolescent mice, in contrast to AMPH (4 mg/kg amphetamine). We did not anticipate that exposure to the ALD in mid-adolescent females would alter *Dcc* expression. We now add data (see Extended data 4) showing that ALD does *not* produce place preference in mid-adolescent female mice, similarly to our observations in early adolescent male mice (Figure 4g). We find it unlikely that ALD treatment would regulate *Dcc* expression in the VTA and we are currently addressing this and similar questions in ongoing studies from the lab aimed at understanding the mechanisms underlying potential beneficial enduring effects of ALD on cognitive maturation (Cuesta et al., *Addiction Biology*).

i) Extended data Figure 4

ii) Extended data Figure 4 legend, Supplemental info page 12:

“(d) ALD does not produce place conditioning in mid-adolescent female mice (Extended data Table 4D).”

iii) Extended data Table 4, Supplemental info page 23:

	Statistical test	Factor	n	Statistic	95% confidence interval	p value (adjusted where appropriate)	corresponding figure
			8/ group				Extended data Figure 4d
D	unpaired t test	Saline vs AMPH CPP		t(14) =1.676		-2.754 to 22.45	0.1159

iv) To further clarify this point, we have now added the following passage to the results section of the manuscript (Page 6):

“The disruptive effects of AMPH on dopamine development in male mice are linked to specific properties of recreational-like doses, which regulate the expression of Dcc in the VTA. While our results suggest that therapeutic-like doses of amphetamine (ALD) do not impact the DA system of mid-adolescent female mice, a full characterization of the sex- and age-dependent effects of this treatment regimen is ongoing.”

5. In Fig. 4k, the NAc varicosities in the saline group looks very different from the varicosities in the saline group in Fig. 4d. How is this explained? Which is more representative? Are tracer efficacies different between the two experiments?

Thank you for pointing this out to us. We want to clarify that we do not make direct comparisons between these two different experiments. This has to do with the sensitivity of the stereological

quantification technique, where differences in stereological counts may arise between experiments where the tissue is processed separately, or where the quantitative analysis is performed by two different individuals. For these reasons, all tissue for one stereological experiment is processed together, and only one experimenter makes all of the counts. Comparisons are then only made within one experiment. For more information on these technical limitations of performing stereology experiments, please see our recent methods chapter (Reynolds et al., Quantifying Dopaminergic Innervation in Rodents Using Unbiased Stereology. in *Dopaminergic System Function and Dysfunction: Experimental Approaches*, 2022).

We do agree with the reviewer that the data of the two saline groups in 4d and 4k appear different from each other. Despite this visual difference, we find no evidence that the tracer efficacies varied between experiments in control groups, as indicated by the similarity in the numbers of eYFP+ varicosities seen in the saline groups of figures 4c and 4j. However, as the tissue of these two experiments was processed separately, and the stereological analyses were conducted by two different researchers in the lab, we cannot compare these two experiments directly.

We now add results showing that the VTA infection levels within the therapeutic-like amphetamine experiment did not differ between the saline and the ALD groups.

i) Extended data figure 4c

ii) Extended data Figure 4 legend, Supplementary information page 12:

“(c) Infection levels of VTA dopamine neurons did not differ between saline and ALD-treated mice (Extended data Table 4C).”

iii) Extended data table 4, Supplementary information page 22:

C			Saline = 3 AMPH = 5				Extended data Figure 4c
	unpaired t test	Saline vs ALD		t(6)= 0.068 31	-1954 to 2066	0.9478	

iv) Figure legend, Figure 4 page 18:

“It is important to note that the visual differences between the photomicrographs in Figures 4b and 4i are due to sampling differences across experiments when taking images and/or to the fact the tissue used in the two experiments was processed separately. However, the basal level of eYFP+ innervation to the PFC likely does not differ between the groups, as they have similar numbers of eYFP+ varicosities in their saline conditions. Comparisons are only made within experiments where all the tissue was processed together and all the quantification was done by a single experimenter.”

6. The DCC overexpression studies are compelling, but are not mimicking “female-like protection” (Abstract). The proposed female protection according to the authors is from Netrin overexpression in the NAc (though this is not proven, as pointed out in #1). It is simply ‘protective’ and should not be compared to the female mechanism. If the authors wish to upregulate Netrin in males to mimic the female pattern, and that confers protection, then this statement would stand. While this reviewer thinks that symmetry with a female vulnerability experiment (#1) would provide the most compelling statement for sexual dimorphism, I will not ask for those studies because a set of rescue studies were performed in males.

This statement has now been modified. Please see the Abstract, page 1 :

*“Upregulating DCC receptor expression in dopamine neurons in adolescent males using a neuron-optimized CRISPR/dCas9 Activation System **protects** against the persistent negative effects of AMPH in early adolescence on adult inhibitory control”*

Please see the response above to Major critique #1 where we provide more information about the additional experiments we now include to address mechanisms underlying differential female vulnerability in response to AMPH.

Minor critiques:

7. Please indicate in the histogram that there is a sex x interaction treatment in Fig. 1j.

We have now updated Fig 1j to clearly indicate the sex by treatment interaction.

8. End of second Results paragraph “These results identify...” is an unnecessary sentence and unlikely to be factually correct. The significance of the paper does not hinge on that sentence.

We have changed the sentence in question (“*These results identify Dcc in the VTA as the first developmental genetic marker shown to be regulated by experience in adolescence in a sexually dimorphic manner*”) to read (Results section, page 3):

“Recent evidence indicates that the expression of homologs of Dcc and its ligand Netrin-1 drive sexual differentiation in c.elegans.^{65,66} Our findings are the first demonstration in mammals that a guidance cue receptor can be regulated in a sexually dimorphic manner in response to the same adolescent experience.”

9. Could you please add more explanation of the intersectional viral tracing technique in Fig. 4 to the main text? I read it in the supplemental methods, but more detail would be nice to better understand the figure.

The main text has now been edited to include the following information in the Results section, page 5:

“To determine the origin of this increase, we used intersectional viral tracing (Fig 4a)⁵³ in male mice exposed to AMPH or to saline in early adolescence to track the growth of dopamine axons as they make targeting decisions at the level of the NAc. To accomplish this, we injected a retrogradely transported Cre-dependent Flp virus in the NAc of DAT^{Cre} mice at PND21, while simultaneously injecting a Flp-dependent eYFP virus at the level of the VTA. This technique limits eYFP expression to dopamine neurons with terminals in the NAc at PND21. We then looked at eYFP+ dopamine axons in the PFC of adult mice, which represent axons of VTA dopamine neurons that were labeled in the NAc at the start of adolescence and which continued to grow to the PFC.”

Reviewer #2 (Remarks to the Author):

General Thoughts:

The manuscript by Reynolds et al., investigates how disruption of DCC/netrin signaling during adolescence impacts dopaminergic axonal growth and behavior. The authors found that

amphetamine during adolescence disrupts axonal outgrowth of DA neurons in males and their performance on a go-no/go task. Females on the other hand are protected against the effects of adolescent amphetamine exposure even though they show changes in *Dcc* and *mir218* after mid-adolescent exposure to amphetamine. The authors use Crisp dCas9 technology to upregulate *Dcc* in the VTA of males during adolescence and show similar effects on DA axonal innervation of the PFC and behavior. These data are exciting and would be impactful but sex-specific effects combined with the male specific endpoints make the manuscript a little disjointed and make it a little difficult to identify a clear take-away.

Comments:

1) The most noteworthy results presented are 1) that females are protected from the effects of adolescent amphetamine exposure, potentially through the concomitant upregulation of netrin protein in the NAC and 2) that upregulation of DCC through the use of Crispr/dCas9 technologies recapitulates the effects of adolescent amphetamine in males.

While both findings are of interest to the field, the female effects are the most novel given that the authors have already published the male specific effects in the past. This study uses new tools to further link DCC to the changes in dopaminergic innervation and behavior. However, there are specific experiments that would make the manuscript very impactful. First, a more thorough investigation of the female specific effects. There are several unanswered questions that need to be addressed. For example, are the same netrin and *dcc* levels observed in females and males. The authors mention that “all dopamine axons in the NAC express DCC” but this work was done in males and given the effects it shouldn’t be assumed that the same expression levels will be observed in females. Without these data the findings are incomplete.

We thank the reviewer for bringing up this important point. We have now added new data and made several changes to the text in order to clarify why we believe that DCC in dopamine neurons plays an important role in both male and female dopamine development.

1) We now explicitly state in the text that *DCC* is enriched in VTA dopamine neurons of both male and female rodents and that the percentage of dopamine neurons expressing *Dcc* does not differ between the sexes. This is evidence published recently from the lab of our collaborator Jeremy Day (Phillips et al., *Cell Reports* 2022). We have also added male and female symbols to the diagram in Figure 1a to clarify that this is the case for both sexes.

i)Figure 1a

ii) Figure 1 Legend, page 12:

(a) *Dcc* mRNA is expressed by 99% of dopamine neurons in the VTA of both male and female mice.⁵⁹

iii) Results section, page 3:

“The guidance cue receptor DCC is highly enriched in dopamine neurons of the ventral tegmental area in both male and female rodents, with no difference between sexes in the percentage of dopamine neurons expressing Dcc mRNA (VTA; Fig 1a,b).”

2) We have now added new neuroanatomical data (Extended data Figure 1) showing that adult female mice have the exact same segregation pattern of DCC protein expression between mesolimbic and mesocortical dopamine axons we have previously reported in adult male mice (Manitt et al., *J Neurosci* 2012). The fact that nucleus accumbens, but not prefrontal cortex, dopamine axons express DCC suggests that the overall role of DCC in segregating mesolimbic and mesocortical pathways is the same regardless of sex. However, we want to emphasize that the timing of this event may differ between the sexes, given that adolescence, and puberty in particular, are not perfectly aligned across chronological age in males and females.

Ongoing studies from our lab are currently focused on fully characterizing male and female dopamine developmental timelines. These studies include detailed assessment of DCC and Netrin-1 expression patterns across normative development. At this point, it would be premature to include the results we have gathered so far in this regard, because we cannot make definitive conclusions yet. In addition, the focus of this paper is the age- and sex-specific regulation of DCC and Netrin-1 in dopamine systems in response to the same adolescent drug experience and these other data will be outside of this main theme.

i) Extended data Figure 1

ii) Extended data Figure 1 Legend, Supplemental material page 9:

Extended data figure 1 *DCC protein is highly expressed in mesolimbic, but not mesocortical dopamine axons of adult female mice.* (a) DCC protein is highly expressed in the NAc of adult female mice, with its expression limited to dopamine axons innervating this region. (b) In the PFC, DCC receptors are expressed by local neurons,¹⁶ but they are rarely or not expressed by dopamine axons. This segregation of DCC expression between the mesolimbic and mesocortical dopamine pathways is identical to what has been previously reported in male mice.¹⁰

iii) We have also added the following information to the Results section, page 3:

“All DCC protein expression in the NAc of female mice is localized to dopamine axons (Extended data Fig 1a). In contrast, few or no dopamine axons in the PFC express DCC receptors (Extended data Fig 1b). The exact same segregation pattern of DCC expression is observed in male mice.⁶⁰”

3) Regarding Netrin-1 upregulation in mid-adolescent females, we have now added additional data to Figure 3 indicating that AMPH-induced upregulation of Netrin-1 in the NAc of mid-adolescent female mice has a protective effect:

i) Figure 3h – We conducted correlational analysis to assess the relationship between *Dcc* mRNA expression in the VTA and Netrin-1 protein levels in the NAc in all of our treatment groups. This analysis revealed a strong and significant *negative* correlation only in the female mice treated with AMPH in mid-adolescence. This result suggests that in mid-adolescent females, drug-induced *downregulation* of *Dcc* mRNA in dopamine axons is tightly associated with the drug-induced *upregulation* of Netrin-1 in the NAc. These results are now included in panel h of Figure 3.

ii) Figure 3 Legend, page 16:

*“(h) *Dcc* mRNA expression in the VTA and Netrin-1 protein levels in the NAc of female mice treated with AMPH in mid-adolescence show a strong and significant negative correlation, suggesting that*

drug-induced downregulation of *Dcc* mRNA in dopamine axons is tightly associated with the drug-induced upregulation of *Netrin-1* in the NAc (Table 3E). “

iii) Detailed statistics Table 3E :

	Statistical test	Factor	n	Statistic	95% confidence interval	p value (adjusted where appropriate)	corresponding figure
E			6				Figure 3h
	Pearson r			$r = -0.8302$, $r^2 = 0.6892$	-0.9809 to -0.05703	0.0408	

iv) There is no correlation between VTA *Dcc* mRNA levels and NAc *Netrin-1* expression in all the other groups. These results are included in Extended Figure 3.

v) Extended data 3 Figure Legend, Supplemental data page 11 :

“(c) There was no correlation between *Netrin-1* protein in the NAc and *Dcc* mRNA in the VTA for mice treated with AMPH or Saline in early adolescence, regardless of sex (Extended data Table 3 A). (d) There was no correlation between *Netrin-1* protein in the NAc and *Dcc* mRNA in the VTA for mid-

adolescent males or female mice treated with saline, nor in mid-adolescent males treated with AMPH (Extended data Table 3 B).”

vi) Extended data Table 3:

	Statistical test	Factor	n	Statistic	95% confidence interval	p value (adjusted where appropriate)	corresponding figure
A							Extended data Figure 3c
	Pearson r	Males + saline	4	r = 0.5482, r ² = 0.3005	-0.8727 to 0.9885	0.4518	
		Females + saline	4	r = -0.1112, r ² = 0.01236	-0.9688 to 0.9516	0.8888	
		Males + AMPH	5	r = -0.04895, r ² = 0.002396	-0.8927 to 0.8709	0.9377	
		Females + AMPH	6	r = 0.5152, r ² = 0.2654	-0.5094 to 0.9356	0.2956	
B							Extended data Figure 3d
	Pearson r	Males + saline	7	r = -0.02616, r ² = 0.0006842	-0.7642 to 0.7415	0.9556	
		Females + saline	6	r = 0.154, r ² = 0.02373	-0.7515 to 0.8583	0.7708	
		Males + AMPH	6	r = 0.2672, r ² = 0.07138	-0.6951 to 0.8865	0.6088	

vii) Figure 3i – To follow up these correlational results, we now add results from a new functional neuroanatomical experiment showing that the upregulation of Netrin-1 in the NAc of female mice treated with AMPH in mid-adolescence indeed protects against the effects of AMPH on dopamine connectivity. To accomplish this, we downregulated *Netrin-1* in the NAc of female mice using shRNA before the AMPH or saline treatment in mid-adolescence. We found that only female mice with *Netrin-1* shRNA + AMPH treatment had an increased expanse of the dopamine input to the PFC in adulthood.

Increased dopamine input volume in the PFC in adulthood is a proxy of ectopic growth of mesolimbic dopamine axons in the PFC. In fact, this finding strongly resembles the effect we have seen in male mice treated with AMPH in early adolescence, which is shown in the current manuscript (Extended data Figure 4a) as well as in our previous work (Reynolds et al., *Neuropsychopharmacology* 2014). These new data support the idea that AMPH-induced upregulation of Netrin-1 in the NAc of mid-adolescent females is protective.

viii) Figure 3 Legend, Page 16 :

*"(i) Top, Netrin-1 protein in the NAc of female mice was downregulated using an shRNA approach before treatment with AMPH or saline in mid-adolescence. Left, Netrin-1 shRNA virus was well expressed in the NAc of female mice. Right, Adult female mice treated with AMPH in mid-adolescence were not different from their saline-treated counterparts when receiving a scrambled control virus, in agreement with the results in Figure 2h. However, adult females had an increased expanse of the dopamine input to the PFC as adults when Netrin-1 in the NAc was downregulated with shRNA before treatment with AMPH in mid-adolescence (Table 3F)."*

ix) Detailed stats table 3 :

| | **Statistical test** | **Factor** | **n** | **Statistic** | **95% confidence interval** | **p value (adjusted where appropriate)** | **corresponding figure** |
|---|---|---|---|---|---|---|---|

F			Scrambled + Saline= 5 Scrambled + AMPH= 5 Netrin-1 shRNA + Saline = 4 Netrin-1 shRNA + AMPH = 5				Figure 3i
	Generalized Estimating Equations analysis (GEE)	Drug (between subjects)		Wald Chi-Square = 2.281 (df = 1)		0.131	
		Virus (between subjects)		Wald Chi-Square = 6.884(df = 1)		0.009	
		Region (within subjects)		Wald Chi-Square = 1546.853 (df = 2)		<0.001	
		Drug x Virus		Wald Chi-Square = 0.589 (df = 1)		0.443	
		Virus x Region		Wald Chi-Square = 2.083 (df = 2)		0.353	
		Drug x Region		Wald Chi-Square = 5.341 (df = 2)		0.69	
		Drug x Virus x Region		Wald Chi-Square = 10.119 (df = 2)		0.006	
	Dunnett's multiple comparisons test	Scrambled:Saline vs. Scrambled:Amphetamine within Cg1		q(15) = 0.9696	-28.73 to 13.19	0.6594	
	Dunnett's multiple comparisons test	Scrambled:Saline vs. Netrin-1 shRNA:Saline within Cg1		q(15) = 1.641	-36.17 to 8.279	0.2763	
	Dunnett's multiple comparisons test	Scrambled:Saline vs. Netrin-1 shRNA:Amphetamine within Cg1		q(15) = 1.537	-33.28 to 8.635	0.323	
	Dunnett's multiple comparisons test	Scrambled:Saline vs. Scrambled:Amphetamine within PrL		q(15) = 0.518	-34.05 to 22.79	0.9182	

	Dunnett's multiple comparisons test	Scrambled:Saline vs. Netrin-1 shRNA:Saline within PrL	q(15) = 0.6796	-37.98 to 22.31	0.8399	
	Dunnett's multiple comparisons test	Scrambled:Saline vs. Netrin-1 shRNA:Amphetamine within PrL	q(15) = 2.746	-58.27 to -1.426	0.0388	
	Dunnett's multiple comparisons test	Scrambled :Saline vs. Scrambled :Amphetamine within IL	q(15) = 0.7311	-9.019 to 16.02	0.8106	
	Dunnett's multiple comparisons test	Scrambled :Saline vs. Netrin-1 shRNA:Saline within IL	q(15) = 0.5618	-16.13 to 10.43	0.8993	
	Dunnett's multiple comparisons test	Scrambled :Saline vs. Netrin-1 shRNA:Amphetamine within IL	q(15) = 2.676	-25.33 to -0.2912	0.0445	

x) We have now made the following modifications to the Results section, page 5:

“In addition, we find a strong, negative correlation between Dcc mRNA in the VTA and Netrin-1 protein in the NAc of females treated with AMPH during mid-adolescence: the greater the downregulation of Dcc levels in the VTA by AMPH, the greater the drug-induced upregulation of Netrin-1 in the NAc (Fig 3h). None of the other groups studied showed a correlation between Dcc in the VTA and Netrin-1 in the NAc (Extended data Fig 3). Therefore, upregulation of Netrin-1 in females may be a compensatory effect of drug treatment, protecting against enduring consequences triggered by drug-induced Dcc downregulation.

To test this idea, we used an shRNA approach to downregulate Netrin-1 expression in the NAc of female mice before subjecting them to AMPH or saline treatment in mid-adolescence (Figure 3i). In adulthood we stereologically quantified the expanse of the dopamine innervation to their PFC. In mid-adolescence, AMPH had no effect on the expanse of the dopamine input to the PFC in mice that received bilateral microinfusion of scrambled shRNA, in agreement with our previous results (Figure 2h). In contrast, AMPH exposure in mid-adolescence produced an increase in the volume of the PFC dopamine input in when it was paired with Netrin-1 downregulation in the NAc (Figure 3i). This increase in PFC dopamine innervation volume mimics our previous results in male mice treated in early adolescence,⁶⁷ providing the first evidence that Netrin-1 upregulation in the NAc of mid-adolescent female mice compensates for the downregulation of Dcc in the VTA and protects females against the deleterious effects of AMPH on adolescent mesocorticolimbic dopamine development.”

2) It seems that the authors are presenting 2 very different stories and therefore need to provide evidence for 2 different sets of conclusions. The male-specific work (Figures 4 & 5) are supported and much of the data presented here replicates work that has been previously published by this group. The female effects need more experiments to fully support their conclusions. The female specific effects described are a rather cursory evaluation of the sex-specific effects of adolescent amphetamine exposure. It is incredibly difficult to tie the two findings together into a complete picture. The authors could use the sex-specific effects to advance our knowledge of how the *DCC system impacts development of the mesocorticolimbic system across the sexes*. For example, the authors could upregulate netrins in the NAC in males and females to see if they replicate the female specific effects and if those effects are protective against early adolescent AMPH exposure in males.

This issue is something we thought about it very carefully when designing and conducting our experiments. After identifying a sex- and an age-specific regulation of *Dcc* by exposure to AMPH, we designed the two lines of follow up experiments to directly assess specific questions within one sex and at each adolescent age group. This is in line with recommendations for the implementation of sex as a biological variable in research (Shansky *Science* 2019), and designed to maximize statistical power while minimizing the number of animals required for the studies.

This important comment has made us realize the need to make modifications to the manuscript to clearly emphasize that the first findings showing a sex- and age-dependent regulation of *Dcc* by AMPH are what made us address the follow up questions in males and females separately.

Indeed, our work yields 3 main conclusions:

First, *Dcc* and *Netrin-1* are differentially regulated in response to the same drug experience in adolescence as a function of specific adolescent treatment age and sex

Second, male mice treated with AMPH in early adolescence show ectopic growth of NAc dopamine axons in the PFC, which depends on the ability of this drug to regulate *Dcc*.

Importantly, this is the first direct demonstration that adolescent experiences can induce long-distance miswiring in brain circuits, and moreover that this occurs in both an age- and sex- dependent manner.

Third, despite a downregulation of *Dcc* by AMPH treatment in mid-adolescence, female mice continue to be protected against deleterious effects via a compensatory upregulation of *Netrin-1*.

We now add data and make changes throughout the manuscript in order to reinforce these three main findings, with the hope that our main message(s) will be conveyed more clearly.

Conclusion 1) Results section, page 3 :

*“Recent evidence indicates that the expression of homologs of *Dcc* and its ligand *Netrin-1* drive sexual differentiation in *c.elegans*.^{65,66} Our findings are the first demonstration in*

mammals that a guidance cue receptor can be regulated in a sexually dimorphic manner in response to the same adolescent experience.”

Conclusion 2)

i) Results section, page 6:

“We find that AMPH in early adolescence also leads to significant restructuring of PFC pyramidal neuron arbors and changes in their spine density in adulthood (Extended data Fig 4 e-i). This effect most likely results from the miswiring of dopamine axons in adolescence, as cell-autonomous manipulation of Dcc levels within dopamine neurons, by altering dopamine innervation to the PFC, substantially shapes the morphology of postsynaptic neurons.^{53,55} Indeed, the miswiring of cortical inputs in early development has been shown to change the organization/function of local cortical networks, making them resemble those of the intended target.⁷³ Our results are the first demonstration that an experience in adolescence produces a long-distance rewiring of the developing brain, leading to enduring alterations to PFC innervation and function. We also show that this event is mediated by sex- and age- specific regulation of guidance cues.”

ii) We have added a paragraph to the discussion to emphasize this important point, page 8-9 :

“Here we identify, for the first time, re-routing of dopaminergic axons from the NAc to the PFC as a sex- and age-dependent response to an adolescent experience, namely exposure to AMPH. This may indicate a male-specific critical period in early adolescence where experiential regulation of Dcc and Netrin-1 can produce cortical miswiring. Understanding the mechanisms by which experiences shape the adolescent brain is still a nascent field, in contrast to well-studied early developmental periods when sensory cortices mature.^{81–86} Cortical miswiring during these early critical periods profoundly shapes the target area, with its network organization more closely resembling that of the axons’ intended target.^{73,87} As the activity patterns and molecular profiles of mesocortical and mesolimbic dopamine axons differ markedly,⁸⁸ how these rerouted connections enduringly impact PFC function and cognitive behaviors are only beginning to be understood.”

Conclusion 3) We have now added additional data to Figure 3 indicating that AMPH-induced upregulation of Netrin-1 in the NAc of mid-adolescent mice has a protective effect:

I) We now explicitly state in the text that DCC is enriched in VTA dopamine neurons of both male and female rodents and that the percentage of dopamine neurons expressing Dcc does not differ between the sexes. This is evidence published recently from the lab of our collaborator Jeremy Day (Phillips et al., *Cell Reports* 2022). We have also added male and female symbols to the diagram in Figure 1a to clarify that this is the case for both sexes.

i)Figure 1a

ii) Figure 1 Legend, page 12:

(a) *Dcc* mRNA is expressed by 99% of dopamine neurons in the VTA of both male and female mice.⁵⁹

iii) Results section, page 3:

“The guidance cue receptor DCC is highly enriched in dopamine neurons of the ventral tegmental area in both male and female rodents, with no difference between sexes in the percentage of dopamine neurons expressing Dcc mRNA (VTA; Fig 1a,b).”

II) We have now added new neuroanatomical data (Extended data Figure 1) showing that adult female mice have the exact same segregation pattern of DCC protein expression between mesolimbic and mesocortical dopamine axons we have previously reported in adult male mice (Manitt et al., *J Neurosci* 2012). The fact that nucleus accumbens, but not prefrontal cortex, dopamine axons express DCC suggests that the overall role of DCC in segregating mesolimbic and mesocortical pathways is the same regardless of sex. However, we want to emphasize that the timing of this event may differ between the sexes, given that adolescence, and puberty in particular, are not perfectly aligned across chronological age in males and females.

Ongoing studies from our lab are currently focused on fully characterizing male and female dopamine developmental timelines. These studies include detailed assessment of DCC and Netrin-1 expression patterns across normative development. At this point, it would be premature to include the results we have gathered so far in this regard, because we cannot make definitive conclusions yet. In addition, the focus of this paper is the age- and sex-specific regulation of DCC and Netrin-1 in dopamine systems in response to the same adolescent drug experience and these other data will be outside of this main theme.

i) Extended data Figure 1

ii) Extended data Figure 1 Legend, Supplemental material page 9:

Extended data figure 1 *DCC protein is highly expressed in mesolimbic, but not mesocortical dopamine axons of adult female mice.* (a) DCC protein is highly expressed in the NAc of adult female mice, with its expression limited to dopamine axons innervating this region. (b) In the PFC, DCC receptors are expressed by local neurons,¹⁶ but they are rarely or not expressed by dopamine axons. This segregation of DCC expression between the mesolimbic and mesocortical dopamine pathways is identical to what has been previously reported in male mice.¹⁰

iii) We have also added the following information to the Results section, page 3:

“All DCC protein expression in the NAc of female mice is localized to dopamine axons (Extended data Fig 1a). In contrast, few or no dopamine axons in the PFC express DCC receptors (Extended data Fig 1b). The exact same segregation pattern of DCC expression is observed in male mice.⁶⁰”

III) Regarding Netrin-1 upregulation in mid-adolescent females, we have now added additional data to Figure 3 indicating that AMPH-induced upregulation of Netrin-1 in the NAc of mid-adolescent female mice has a protective effect:

i) Figure 3h – We conducted correlational analysis to assess the relationship between *Dcc* mRNA expression in the VTA and Netrin-1 protein levels in the NAc in all of our treatment groups. This analysis revealed a strong and significant *negative* correlation only in the female mice treated with AMPH in mid-adolescence. This result suggests that in mid-adolescent females, drug-induced *downregulation* of *Dcc* mRNA in dopamine axons is tightly associated with the drug-induced *upregulation* of Netrin-1 in the NAc. These results are now included in panel h of Figure 3.

ii) Figure 3 Legend, page 16:

*“(h) *Dcc* mRNA expression in the VTA and Netrin-1 protein levels in the NAc of female mice treated with AMPH in mid-adolescence show a strong and significant negative correlation, suggesting that drug-induced downregulation of *Dcc* mRNA in dopamine axons is tightly associated with the drug-induced upregulation of Netrin-1 in the NAc (Table 3E).”*

iii) Detailed statistics Table 3E :

	Statistical test	Factor	n	Statistic	95% confidence interval	p value (adjusted where appropriate)	corresponding figure
E			6				Figure 3h
	Pearson r			$r = -0.8302,$ $r^2 = 0.6892$	-0.9809 to - 0.05703	0.0408	

iv) There is no correlation between VTA *Dcc* mRNA levels and NAc Netrin-1 expression in all the other groups. These results are included in Extended Figure 3.

v) Extended data 3 Figure Legend, Supplemental data page 11 :

“(c) There was no correlation between Netrin-1 protein in the NAc and Dcc mRNA in the VTA for mice treated with AMPH or Saline in early adolescence, regardless of sex (Extended data Table 3 A). (d) There was no correlation between Netrin-1 protein in the NAc and Dcc mRNA in the VTA for mid-adolescent males or female mice treated with saline, nor in mid-adolescent males treated with AMPH (Extended data Table 3 B).”

vi) Extended data Table 3:

	Statistical test	Factor	n	Statistic	95% confidence interval	p value (adjusted where appropriate)	corresponding figure
A							Extended data Figure 3c
	Pearson r	Males + saline	4	r = 0.5482, r2 = 0.3005	-0.8727 to 0.9885	0.4518	
		Females + saline	4	r = -0.1112, r2 = 0.01236	-0.9688 to 0.9516	0.8888	

		Males + AMPH	5	r = -0.04895, r2 = 0.002396	-0.8927 to 0.8709	0.9377	
		Females + AMPH	6	r = 0.5152, r2 = 0.2654	-0.5094 to 0.9356	0.2956	
B							Extended data Figure 3d
	Pearson r	Males + saline	7	r = -0.02616, r2 = 0.0006842	-0.7642 to 0.7415	0.9556	
		Females + saline	6	r = 0.154, r2 = 0.02373	-0.7515 to 0.8583	0.7708	
		Males + AMPH	6	r = 0.2672, r2 = 0.07138	-0.6951 to 0.8865	0.6088	

vii) Figure 3i – To follow up these correlational results, we now add results from a new functional neuroanatomical experiment showing that the upregulation of Netrin-1 in the NAc of female mice treated with AMPH in mid-adolescence indeed protects against the effects of AMPH on dopamine connectivity. To accomplish this, we downregulated *Netrin-1* in the NAc of female mice using shRNA before the AMPH or saline treatment in mid-adolescence. We found that only female mice with *Netrin-1* shRNA + AMPH treatment had an increased expanse of the dopamine input to the PFC in adulthood.

Increased dopamine input volume in the PFC in adulthood is a proxy of ectopic growth of mesolimbic dopamine axons in the PFC. In fact, this finding strongly resembles the effect we have seen in male mice treated with AMPH in early adolescence, which is shown in the current manuscript (Extended data Figure 4a) as well as in our previous work (Reynolds et al., *Neuropsychopharmacology* 2014). These new data support the idea that AMPH-induced upregulation of Netrin-1 in the NAc of mid-adolescent females is protective.

F			Scrambled + Saline= 5 Scrambled + AMPH= 5 Netrin-1 shRNA + Saline = 4 Netrin-1 shRNA + AMPH = 5				Figure 3i
	Generalized Estimating Equations analysis (GEE)	Drug (between subjects)		Wald Chi-Square = 2.281 (df = 1)		0.131	
		Virus (between subjects)		Wald Chi-Square = 6.884(df = 1)		0.009	
		Region (within subjects)		Wald Chi-Square = 1546.853 (df = 2)		<0.001	
		Drug x Virus		Wald Chi-Square = 0.589 (df = 1)		0.443	
		Virus x Region		Wald Chi-Square = 2.083 (df = 2)		0.353	
		Drug x Region		Wald Chi-Square = 5.341 (df = 2)		0.69	
		Drug x Virus x Region		Wald Chi-Square = 10.119 (df = 2)		0.006	
	Dunnett's multiple comparisons test	Scrambled:Saline vs. Scrambled:Amphetamine within Cg1		q(15) = 0.9696	-28.73 to 13.19	0.6594	
	Dunnett's multiple comparisons test	Scrambled:Saline vs. Netrin-1 shRNA:Saline within Cg1		q(15) = 1.641	-36.17 to 8.279	0.2763	
	Dunnett's multiple comparisons test	Scrambled:Saline vs. Netrin-1 shRNA:Amphetamine within Cg1		q(15) = 1.537	-33.28 to 8.635	0.323	
	Dunnett's multiple comparisons test	Scrambled:Saline vs. Scrambled:Amphetamine within PrL		q(15) = 0.518	-34.05 to 22.79	0.9182	

	Dunnett's multiple comparisons test	Scrambled:Saline vs. Netrin-1 shRNA:Saline within PrL	q(15) = 0.6796	-37.98 to 22.31	0.8399	
	Dunnett's multiple comparisons test	Scrambled:Saline vs. Netrin-1 shRNA:Amphetamine within PrL	q(15) = 2.746	-58.27 to -1.426	0.0388	
	Dunnett's multiple comparisons test	Scrambled :Saline vs. Scrambled :Amphetamine within IL	q(15) = 0.7311	-9.019 to 16.02	0.8106	
	Dunnett's multiple comparisons test	Scrambled :Saline vs. Netrin-1 shRNA:Saline within IL	q(15) = 0.5618	-16.13 to 10.43	0.8993	
	Dunnett's multiple comparisons test	Scrambled :Saline vs. Netrin-1 shRNA:Amphetamine within IL	q(15) = 2.676	-25.33 to -0.2912	0.0445	

x) We have now made the following modifications to the Results section, page 5:

“In addition, we find a strong, negative correlation between Dcc mRNA in the VTA and Netrin-1 protein in the NAc of females treated with AMPH during mid-adolescence: the greater the downregulation of Dcc levels in the VTA by AMPH, the greater the drug-induced upregulation of Netrin-1 in the NAc (Fig 3h). None of the other groups studied showed a correlation between Dcc in the VTA and Netrin-1 in the NAc (Extended data Fig 3). Therefore, upregulation of Netrin-1 in females may be a compensatory effect of drug treatment, protecting against enduring consequences triggered by drug-induced Dcc downregulation.

To test this idea, we used an shRNA approach to downregulate Netrin-1 expression in the NAc of female mice before subjecting them to AMPH or saline treatment in mid-adolescence (Figure 3i). In adulthood we stereologically quantified the expanse of the dopamine innervation to their PFC. In mid-adolescence, AMPH had no effect on the expanse of the dopamine input to the PFC in mice that received bilateral microinfusion of scrambled shRNA, in agreement with our previous results (Figure 2h). In contrast, AMPH exposure in mid-adolescence produced an increase in the volume of the PFC dopamine input in when it was paired with Netrin-1 downregulation in the NAc (Figure 3i). This increase in PFC dopamine innervation volume mimics our previous results in male mice treated in early adolescence,⁶⁷ providing the first evidence that Netrin-1 upregulation in the NAc of mid-adolescent female mice compensates for the downregulation of Dcc in the VTA and protects females against the deleterious effects of AMPH on adolescent mesocorticolimbic dopamine development.”

3) The data analysis appears correct but there are some flaws in the interpretation and conclusions. The biggest flaw is in the assumption that DCC and netrin in dopamine neurons is the same across the sexes. The studies investigating DCC in dopamine neurons were done in males (reference #s 61 & 62). The authors show here that DCC/netrin effects are sex-specific but assume that the expression patterns will be the same. This is not necessarily the case. It would be helpful for the authors to do a full assessment of DCC and netrin in females before they make claims about the mechanisms regulating protection from AMPH exposure in females. The conclusions and interpretation should at least be revised to reflect that lack of studies in females and that many of the interpretations are based on findings that have only been done in males.

An example of this assumption is provided here:

“In the PFC, mNetrin-1 levels are high but only few dopamine axons express DCC. In contrast, Netrin-1 levels are lower in the nucleus accumbens (NAc), but all dopamine axons in this region highly express DCC (Fig 3a).⁶⁶ Furthermore, dopamine axons are the only source of DCC protein expression in the NAc,⁶⁶ suggesting a crucial and complementary role of DCC in dopamine axons and Netrin-1 in the NAc in the developmental organization of mesocorticolimbic dopamine connectivity. Finally, Netrin-1 levels in the NAc decline across adolescence in male mice,⁶¹”

These two studies were only done in males. It is unclear that the authors can assume these would be the same in females, especially given the sex-specific effects already observed.

We agree with the reviewer that the initial work referenced in this paper was done only in male rodents. We now add new data from experiments performed in female mice and provide information regarding DCC receptor expression and the potential protective role of Netrin-1 upregulation in the nucleus accumbens against AMPH-induced dopamine development disruption. Below we indicate the changes we made throughout the manuscript.

1.) We have now added new data and made several changes to the text in order to clarify why we believe that DCC in dopamine neurons plays an important role in both male and female dopamine development.

I) We now explicitly state in the text that *DCC* is enriched in VTA dopamine neurons of both male and female rodents and that the percentage of dopamine neurons expressing *Dcc* does not differ between the sexes. This is evidence published recently from the lab of our collaborator Jeremy Day (Phillips et al., *Cell Reports* 2022). We have also added male and female symbols to the diagram in Figure 1a to clarify that this is the case for both sexes.

i) Figure 1a

ii) Figure 1 Legend, page 12:

(a) *Dcc* mRNA is expressed by 99% of dopamine neurons in the VTA of both male and female mice.⁵⁹

iii) Results section, page 3:

“The guidance cue receptor DCC is highly enriched in dopamine neurons of the ventral tegmental area in both male and female rodents, with no difference between sexes in the percentage of dopamine neurons expressing Dcc mRNA (VTA; Fig 1a,b).”

II) We have now added new neuroanatomical data (Extended data Figure 1) showing that adult female mice have the exact same segregation pattern of DCC protein expression between mesolimbic and mesocortical dopamine axons we have previously reported in adult male mice (Manitt et al., *J Neurosci* 2012). The fact that nucleus accumbens, but not prefrontal cortex, dopamine axons express DCC suggests that the overall role of DCC in segregating mesolimbic and mesocortical pathways is the same regardless of sex. However, we want to emphasize that the timing of this event may differ between the sexes, given that adolescence, and puberty in particular, are not perfectly aligned across chronological age in males and females.

Ongoing studies from our lab are currently focused on fully characterizing male and female dopamine developmental timelines. These studies include detailed assessment of DCC and Netrin-1 expression patterns across normative development. At this point, it would be premature to include the results we have gathered so far in this regard, because we cannot make definitive conclusions yet. In addition, the focus of this paper is the age- and sex-specific regulation of DCC and Netrin-1 in dopamine systems in response to the same adolescent drug experience and these other data will be outside of this main theme.

i) Extended data Figure 1

ii) Extended data Figure 1 Legend, Supplemental material page 9:

Extended data figure 1 *DCC protein is highly expressed in mesolimbic, but not mesocortical dopamine axons of adult female mice.* (a) DCC protein is highly expressed in the NAc of adult female mice, with its expression limited to dopamine axons innervating this region. (b) In the PFC, DCC receptors are expressed by local neurons,¹⁶ but they are rarely or not expressed by dopamine axons. This segregation of DCC expression between the mesolimbic and mesocortical dopamine pathways is identical to what has been previously reported in male mice.¹⁰

iii) We have also added the following information to the Results section, page 3:

“All DCC protein expression in the NAc of female mice is localized to dopamine axons (Extended data Fig 1a). In contrast, few or no dopamine axons in the PFC express DCC receptors (Extended data Fig 1b). The exact same segregation pattern of DCC expression is observed in male mice.⁶⁰”

III) Regarding Netrin-1 upregulation in mid-adolescent females, we have now added additional data to Figure 3 indicating that AMPH-induced upregulation of Netrin-1 in the NAc of mid-adolescent female mice has a protective effect:

i) Figure 3h – We conducted correlational analysis to assess the relationship between *Dcc* mRNA expression in the VTA and Netrin-1 protein levels in the NAc in all of our treatment groups. This analysis revealed a strong and significant *negative* correlation only in the female mice treated with AMPH in mid-adolescence. This result suggests that in mid-adolescent females, drug-induced *downregulation* of *Dcc* mRNA in dopamine axons is tightly associated with the drug-induced *upregulation* of Netrin-1 in the NAc. These results are now included in panel h of Figure 3.

ii) Figure 3 Legend, page 16:

*“(h) *Dcc* mRNA expression in the VTA and Netrin-1 protein levels in the NAc of female mice treated with AMPH in mid-adolescence show a strong and significant negative correlation, suggesting that drug-induced downregulation of *Dcc* mRNA in dopamine axons is tightly associated with the drug-induced upregulation of Netrin-1 in the NAc (Table 3E).”*

iii) Detailed statistics Table 3E :

	Statistical test	Factor	n	Statistic	95% confidence interval	p value (adjusted where appropriate)	corresponding figure
E			6				Figure 3h
	Pearson r			$r = -0.8302,$ $r^2 = 0.6892$	-0.9809 to - 0.05703	0.0408	

iv) There is no correlation between VTA *Dcc* mRNA levels and NAc Netrin-1 expression in all the other groups. These results are included in Extended Figure 3.

v) Extended data 3 Figure Legend, Supplemental data page 11 :

“(c) There was no correlation between Netrin-1 protein in the NAc and Dcc mRNA in the VTA for mice treated with AMPH or Saline in early adolescence, regardless of sex (Extended data Table 3 A). (d) There was no correlation between Netrin-1 protein in the NAc and Dcc mRNA in the VTA for mid-adolescent males or female mice treated with saline, nor in mid-adolescent males treated with AMPH (Extended data Table 3 B).”

vi) Extended data Table 3:

	Statistical test	Factor	n	Statistic	95% confidence interval	p value (adjusted where appropriate)	corresponding figure
A							Extended data Figure 3c
	Pearson r	Males + saline	4	$r = 0.5482$, $r^2 = 0.3005$	-0.8727 to 0.9885	0.4518	
		Females + saline	4	$r = -0.1112$, $r^2 = 0.01236$	-0.9688 to 0.9516	0.8888	

		Males + AMPH	5	r = -0.04895, r2 = 0.002396	-0.8927 to 0.8709	0.9377	
		Females + AMPH	6	r = 0.5152, r2 = 0.2654	-0.5094 to 0.9356	0.2956	
B							Extended data Figure 3d
	Pearson r	Males + saline	7	r = -0.02616, r2 = 0.0006842	-0.7642 to 0.7415	0.9556	
		Females + saline	6	r = 0.154, r2 = 0.02373	-0.7515 to 0.8583	0.7708	
		Males + AMPH	6	r = 0.2672, r2 = 0.07138	-0.6951 to 0.8865	0.6088	

vii) Figure 3i – To follow up these correlational results, we now add results from a new functional neuroanatomical experiment showing that the upregulation of Netrin-1 in the NAc of female mice treated with AMPH in mid-adolescence indeed protects against the effects of AMPH on dopamine connectivity. To accomplish this, we downregulated *Netrin-1* in the NAc of female mice using shRNA before the AMPH or saline treatment in mid-adolescence. We found that only female mice with *Netrin-1* shRNA + AMPH treatment had an increased expanse of the dopamine input to the PFC in adulthood.

Increased dopamine input volume in the PFC in adulthood is a proxy of ectopic growth of mesolimbic dopamine axons in the PFC. In fact, this finding strongly resembles the effect we have seen in male mice treated with AMPH in early adolescence, which is shown in the current manuscript (Extended data Figure 4a) as well as in our previous work (Reynolds et al., *Neuropsychopharmacology* 2014). These new data support the idea that AMPH-induced upregulation of Netrin-1 in the NAc of mid-adolescent females is protective.

F			Scrambled + Saline= 5 Scrambled + AMPH= 5 Netrin-1 shRNA + Saline = 4 Netrin-1 shRNA + AMPH = 5				Figure 3i
	Generalized Estimating Equations analysis (GEE)	Drug (between subjects)		Wald Chi-Square = 2.281 (df = 1)		0.131	
		Virus (between subjects)		Wald Chi-Square = 6.884(df = 1)		0.009	
		Region (within subjects)		Wald Chi-Square = 1546.853 (df = 2)		<0.001	
		Drug x Virus		Wald Chi-Square = 0.589 (df = 1)		0.443	
		Virus x Region		Wald Chi-Square = 2.083 (df = 2)		0.353	
		Drug x Region		Wald Chi-Square = 5.341 (df = 2)		0.69	
		Drug x Virus x Region		Wald Chi-Square = 10.119 (df = 2)		0.006	
	Dunnett's multiple comparisons test	Scrambled:Saline vs. Scrambled:Amphetamine within Cg1		q(15) = 0.9696	-28.73 to 13.19	0.6594	
	Dunnett's multiple comparisons test	Scrambled:Saline vs. Netrin-1 shRNA:Saline within Cg1		q(15) = 1.641	-36.17 to 8.279	0.2763	
	Dunnett's multiple comparisons test	Scrambled:Saline vs. Netrin-1 shRNA:Amphetamine within Cg1		q(15) = 1.537	-33.28 to 8.635	0.323	
	Dunnett's multiple comparisons test	Scrambled:Saline vs. Scrambled:Amphetamine within PrL		q(15) = 0.518	-34.05 to 22.79	0.9182	

	Dunnett's multiple comparisons test	Scrambled:Saline vs. Netrin-1 shRNA:Saline within PrL	q(15) = 0.6796	-37.98 to 22.31	0.8399	
	Dunnett's multiple comparisons test	Scrambled:Saline vs. Netrin-1 shRNA:Amphetamine within PrL	q(15) = 2.746	-58.27 to -1.426	0.0388	
	Dunnett's multiple comparisons test	Scrambled :Saline vs. Scrambled :Amphetamine within IL	q(15) = 0.7311	-9.019 to 16.02	0.8106	
	Dunnett's multiple comparisons test	Scrambled :Saline vs. Netrin-1 shRNA:Saline within IL	q(15) = 0.5618	-16.13 to 10.43	0.8993	
	Dunnett's multiple comparisons test	Scrambled :Saline vs. Netrin-1 shRNA:Amphetamine within IL	q(15) = 2.676	-25.33 to -0.2912	0.0445	

x) We have now made the following modifications to the Results section, page 5:

“In addition, we find a strong, negative correlation between Dcc mRNA in the VTA and Netrin-1 protein in the NAc of females treated with AMPH during mid-adolescence: the greater the downregulation of Dcc levels in the VTA by AMPH, the greater the drug-induced upregulation of Netrin-1 in the NAc (Fig 3h). None of the other groups studied showed a correlation between Dcc in the VTA and Netrin-1 in the NAc (Extended data Fig 3). Therefore, upregulation of Netrin-1 in females may be a compensatory effect of drug treatment, protecting against enduring consequences triggered by drug-induced Dcc downregulation.

To test this idea, we used an shRNA approach to downregulate Netrin-1 expression in the NAc of female mice before subjecting them to AMPH or saline treatment in mid-adolescence (Figure 3i). In adulthood we stereologically quantified the expanse of the dopamine innervation to their PFC. In mid-adolescence, AMPH had no effect on the expanse of the dopamine input to the PFC in mice that received bilateral microinfusion of scrambled shRNA, in agreement with our previous results (Figure 2h). In contrast, AMPH exposure in mid-adolescence produced an increase in the volume of the PFC dopamine input in when it was paired with Netrin-1 downregulation in the NAc (Figure 3i). This increase in PFC dopamine innervation volume mimics our previous results in male mice treated in early adolescence,⁶⁷ providing the first evidence that Netrin-1 upregulation in the NAc of mid-adolescent female mice compensates for the downregulation of Dcc in the VTA and protects females against the deleterious effects of AMPH on adolescent mesocorticolimbic dopamine development.”

2.) We make changes to the text throughout the manuscript to clarify which previous results were known only in male animals.

i) Results section, page 3:

“Dopaminergic axons in the NAc of adult female mice strongly express DCC protein (Extended data Fig 1a), while few PFC dopamine axons co-localize with DCC (Extended data Fig 1b), a pattern identical to what has been observed in adult male mice.⁶⁰ The expression of DCC protein and Dcc mRNA in the VTA decreases across postnatal development,^{58,61,62} and can be altered by experience at discrete time points in male animals. However, whether the effects of experience on Dcc expression are both age- and sex-specific remains to be explored.”

ii) Results section, page 4 :

“In male mice, Netrin-1 is expressed in the terminal regions of the mesocorticolimbic dopamine system, albeit in a complementary manner to the expression levels of DCC receptors in the dopamine axons that innervate these areas. In the PFC, Netrin-1 levels are high but only few dopamine axons express DCC. In contrast, Netrin-1 levels are lower in the nucleus accumbens (NAc), but all dopamine axons in this region highly express DCC (Fig 3a).⁶⁰ Netrin-1 levels in the NAc decline across adolescence in male mice,⁶² mirroring the same developmental pattern as we see in Dcc expression.²³⁻²⁵ Furthermore, dopamine axons are the only source of DCC protein expression in the NAc of adult male mice,⁶⁰ suggesting a crucial and complementary role of DCC in dopamine axons and Netrin-1 in the NAc in the developmental organization of mesocorticolimbic dopamine connectivity. We have recently found that this exact same pattern of DCC expression is present in adult female mice, with dopamine axons in the NAc heavily expressing DCC, and PFC dopamine axons rarely co-localizing with DCC (Extended data Fig 1).”

iii) Discussion section, page 9 :

“An important point to highlight is that the vast majority of previous studies of the role of the Netrin-1/DCC system in dopamine development and in axon pathfinding were performed only in male subjects. The current study represents only a first step toward unraveling how sex influences the expression and function of this guidance cue system.”

3.) These previous data referenced in this manuscript were derived from a large set of studies from our lab using male mice only. Several ongoing studies from our team are aimed at (i) fully characterizing differences or similarities in the developmental trajectories of Netrin-1 and Dcc expression in dopamine systems in males and females and (ii) understanding the function of this guidance cue system in normative mesocorticolimbic dopamine development across both sexes. This is a large endeavour and involves longitudinal experiments, assessment of transcript and protein expression in different brain regions and using various methods, as well as gain and loss of function experiments in both sexes, using many experimental groups. Our initial results are very intriguing, but

this work is not completed and at this point we cannot draw definitive conclusions. It would be premature to add only a part of these data to this manuscript. In addition, we believe that a full characterization of the expression and function of the Netrin-1/DCC system in normative dopamine development in males and female mice would fall outside the scope of the current manuscript, which is that a guidance cue system can be differentially regulated in an age- and sex-specific manner in response to the same experience, and that AMPH in early adolescent male mice triggers ectopic innervation of dopamine axons to the PFC.

4) Generally speaking, the methodology is seemingly sound. However, there are a few questions that need to be addressed to help assess this. First, in Figure 4g, can the authors please explain what the “delta place preference” means? One would assume the author showing the difference between a pretest “preference” and a posttest preference. However, the units don’t necessarily make sense with that calculation. Is this a differences in percent time?

We would like to thank the reviewer for bringing our attention to this error. The change in place preference is measured as a difference in the percentage of time spent in the paired chamber between the post-test and pre-test. The axis for figure 4g has been updated accordingly to read “*Δ place preference (%)*”. In addition, we have clarified this issue in the Methods (Supplementary information, page 6) :

*“A delta preference score was then calculated for each mouse by subtracting the **percentage of time spent in the originally unpreferred chamber during the pretest** from the **percentage of time spent in that same chamber during the post-test**, Place Preference = % time POST - % time PRE.”*

5) Additionally, in figure 4, can the authors please explain why there appears to be such big differences in their saline animals between figure 4b and 4i?

Thank you for pointing this out to us. We want to clarify that we do not make direct comparisons between these two different experiments. This has to do with the sensitivity of the stereological quantification technique, where differences in stereological counts may arise between experiments where the tissue is processed separately, or where the quantitative analysis is performed by two different individuals. For these reasons, all tissue for one stereological experiment is processed together, and only one experimenter makes all of the counts. Comparisons are then only made within one experiment. For more information on these technical limitations of performing stereology experiments, please see our recent methods chapter (Reynolds et al., Quantifying Dopaminergic Innervation in Rodents Using Unbiased Stereology. in *Dopaminergic System Function and Dysfunction: Experimental Approaches*, 2022).

We do agree with the reviewer that the data of the two saline groups in 4d and 4k appear different from each other. Despite this visual difference, we find no evidence that the tracer efficacies varied between experiments in control groups, as indicated by the similarity in the numbers of eYFP+ varicosities seen in the saline groups of figures 4c and 4j. However, as the tissue of these two

experiments was processed separately, and the stereological analyses were conducted by two different researchers in the lab, we cannot compare these two experiments directly.

We now add results showing that the VTA infection levels within the therapeutic-like amphetamine experiment did not differ between the saline and the ALD groups.

i) Extended data figure 4c

ii) Extended data Figure 4 legend, Supplementary information page 12:

“(c) Infection levels of VTA dopamine neurons did not differ between saline and ALD-treated mice (Extended data Table 4C).”

iii) Extended data table 4, Supplementary information page 22:

C			Saline = 3 AMPH = 5				Extended data Figure 4c
	unpaired t test	Saline vs ALD		t(6)= 0.068 31	-1954 to 2066	0.9478	

iv) Figure legend, Figure 4 page 18:

“It is important to note that the visual differences between the photomicrographs in Figures 4b and 4i are due to sampling differences across experiments when taking images and/or to the fact the tissue used in the two experiments was processed separately. However, the basal level of eYFP+ innervation to the PFC likely does not differ between the groups, as they have similar numbers of eYFP+ varicosities in their saline conditions. Comparisons are only made within experiments where all the tissue was processed together and all the quantification was done by a single experimenter.”

6) On the stats table, the last figure is mislabeled – it should be figure 5 but is labeled as figure 2.

This has now been corrected in the detailed stats table for Figure 5. We thank the reviewer for bringing this typographical error to our attention.

Reviewer #3 (Remarks to the Author):

This is an excellent manuscript describing a dcc-netrin mechanism by which early adolescent amphetamine (AMPH) exposure causes VTA dopaminergic neurons to ectopically target the PFC instead of the NAcc in males, with consequential deficits in adult impulse responsivity. Interestingly, females show reduced dcc in the VTA following late adolescent, rather than early adolescent AMPH exposure, but this did not translate to impulsivity in adulthood. This was attributed to increased nestin expression in the NAcc, which compensated for decreased dcc and allowed proper targeting of DA neurons to the NAcc. Methodology used was innovative and robust, and data was analyzed and interpreted appropriately. The methods in the expanded section were sufficiently detailed (I assume that inclusion in the main manuscript is not required, though this is an editorial question). The results are highly significant, as they are the first to demonstrate, using a molecular- region-, and time-specific prevention of a behavior that is strongly predictive of drug addiction following adolescent drug exposure, and clearly demonstrates a mechanism underpinning male vulnerability to drug addiction following adolescent exposure as well. I did not find much to substantively critique about these studies or the report. There were, however, several unfortunate groups/experiments lacking which would have completed the story of sex-specificity, and a few other minor issues that if addressed could improve the overall manuscript.

1. Figure 1i: The graph illustrates and results section reports that miR-218 is negatively correlated with Dcc mRNA in females, but the figure caption states that they were “positively associated”

We thank the reviewer for pointing out this error. The figure legend has been updated as follows :

*“In early adolescence, VTA miR-218 and Dcc mRNA levels correlated **negatively** in male, (f) but not female mice (g) (Table 1C,D).”*

And :

*“In mid-adolescence, VTA miR-218 and Dcc mRNA levels did not correlate in males (k) but were **negatively correlated** in females (l) (Table 1G,H).”*

2. Figure 2 and results: While it is appreciated that previous work by this group (and figure 5 of this paper) showed male impulsivity following adolescent amphetamine, reporting a sex-specific effect of adolescent amphetamine on enduring behavior requires a simultaneous demonstration of male/female data in the same experiment with the same strain of mice, and for sex to be analyzed as a biological variable (as done in the other experiments reported in the manuscript).

Thank you for this comment. This issue is something we thought about it very carefully when designing and conducting our experiments. After identifying a sex- and an age-specific regulation of *Dcc* by exposure to AMPH, we designed the two lines of follow up experiments to directly assess specific questions within one sex and at each adolescent age group. This is in line with recommendations for the implementation of sex as a biological variable in research (Shansky *Science* 2019), and designed to maximize statistical power while minimizing the number of animals required for the studies. For the behavioral data in particular, (i) we have already demonstrated the deleterious effect of AMPH exposure in early adolescence on inhibitory control in adulthood (Reynolds et al., *Biol Psych* 2018), (ii) we replicate this result in the CRISPRa experiments shown in this manuscript as can be seen in Figure 5. Indeed, male mice that received bilateral infection of the LacZ sgRNA construct and AMPH treatment in early adolescence (*LacZ* sgRNA + AMPH) show impaired inhibitory control in adulthood compared to their saline counterparts (*LacZ* sgRNA + Saline - Figure 5-panel m).

3. Figure 4: It is not clear to me that ventral PFC regions were observed to be more innervated by TH+ terminals in the extended data (there does not seem to be a statistical subregion effect), as suggested in the results (when it is stated that the YFP+ results were consistent with overall DA innervation).

This is an important point. Yes, there is no treatment x subregion interaction in the PFC TH+ volume (Extended data Figure 4a) and we have now reorganized and modified information pertaining the TH+ volume data presented in the Extended data. Please see the Results section, page 6 :

“We found significantly more eYFP+ dopamine axon terminals in the PFC of adult mice that were exposed to AMPH in early adolescence, in comparison to saline-exposed counterparts (Fig 4b). This increase is in line with the overall changes in dopamine input volume seen in the same brain sections (Extended data Fig 4a) and previously reported.⁶⁷ The AMPH-induced increase in eYFP+ terminals in the PFC was more pronounced in ventral subregions (Fig 4c).”

4. Similar to point 2, it is unfortunate that female DA innervation of NAcc and PFC were not reported; although the behavioral and *dcc*/nestin data suggest that innervation is not altered in females, this remains a question unless it is measured.

We did report data about the DA innervation in the PFC in the female mice exposed to AMPH or to Saline in early or in mid-adolescence. We now realize that we needed to highlight this information more clearly and have made changes to address this important point.

We now make changes to Figure 2 to clarify the information presented including :

i) Fig 2b – explicit labeling of the two experiments performed above their diagrams

ii) Fig 2c and 2h – addition of a “PFC Stereology” label above the results from the stereology experiments.

iii) Fig 2d-g and 2i-l – addition of a label of “Action impulsivity” to clearly identify all results from the Go/No-Go task, as we had already in Figure 5.

We do not analyze NAc dopamine innervation in the female mice in this paper as i) we found no changes in the PFC dopamine innervation, and ii) even when dramatic changes in PFC dopamine innervation are apparent, we are not always able to capture these differences in the NAc. This is because the dopamine innervation to the NAc is ~40-fold greater than the innervation to the PFC (Manitt et al., *Trans Psych* 2013, and as such it is difficult to capture changes in NAc dopamine circuitry using stereology to assess only TH+ varicosity counts as the change we see in the PFC varicosity number is well below the standard error for counts in the NAc. In male mice we have found that increases to eYFP-labeled dopamine innervation to the PFC are strongly associated with a reduction in eYFP-labeled dopamine terminals in the NAc (Figure 4c-e), but this effect in the NAc is only apparent when considering the subset of eYFP-labeled terminals as otherwise it falls within the statistical error of the TH+ innervation.

5. The data (YFP+ innervation of PFC, and negative correlation between NAc and PFC terminals) is fascinating. Do the authors believe that these axons had once reached the NAc in early adolescence (in order to be retrogradely labeled) and then rerouted to the PFC during adolescence? Has this ever been seen before, i.e., rather than a mis-direction, there is actually a re-direction once a target is already reached? A discussion would be interesting about the potential time course of possibilities for intervention while the axons are moving to a new destination.

Thank you for raising these interesting points. We agree that this result is quite interesting and important. We have pointed at this process before, when we first discovered that dopamine axons are still growing from the NAc to the PFC across adolescence (see Reynolds et al 2018). Indeed, DA neurons have extended their axons to the NAc by early adolescence and do not require DCC receptors to reach this target. We believe that axons innervating the PFC also arrive early in the NAc and pause

there before continuing their pathway to the PFC. This is something previously described for other neurotransmitter pathways during embryonic development.

We now include additional text to the Discussion section where we discuss this idea, please see the Discussion section, page 9 :

“Our findings provide important mechanistic insight regarding the critical role for the Netrin-1/DCC system in adolescent neurodevelopment,^{4,13} and its strong link to psychiatric disorders of an adolescent onset.^{96,97} DCC receptors are not required for DA axons to grow to the NAc, as even mice with a homozygous deletion of DCC in DA neurons show DA innervation to the NAc.⁵⁵ As axons from DA neurons extend to reach anterior regions, they pass through intermediate targets along their route. The NAc is a particularly interesting structure, because it appears to be a choice point where a large number of DA axons establish their final connections whereas it serves merely as a waypoint for axons extending to the PFC. DCC expression in DA axons in adolescence determines whether DA axons recognize the NAc as their final target or continuing to grow to the PFC.⁵³ It is likely that mesocortical DA axons arrive early in the NAc and pause before continuing their journey to the PFC. Indeed, DA axons have been shown to pause at intermediate pathfinding points early in embryonic development,⁹⁸⁻¹⁰⁰ and guidance cues have been shown to orchestrate waiting periods in corticothalamic axon pathfinding.¹⁰¹ Alternatively mesocortical DA axons may slowly but continuously extend from the VTA throughout adolescence.”

6. Figure 5: It appears that the lack of difference between Sal-treated and AMPH-treated dcc overexpressing mice was due to a “meeting in the middle” between lacZ sgRNA SAL- and AMPH-treated animals, whereby SAL-treated dcc-overexpressing mice showed higher impulsivity. The authors should address this.

Since upregulation of DCC receptors in itself alters the development of DA neurons in adolescence, we are not surprised to find that adult Saline-treated mice with LacZ and with *Dcc*-sgRNA show a different number of commission errors in the Go/No-Go task. We believe that altering *Dcc* expression, even by increasing it, may alter dopaminergic development. Importantly, there is not a significant difference in the area under the curve (AUC) between these two saline groups, unlike between the saline and AMPH treated LacZ controls. This indicates that while the pattern of their responses over time may differ, the overall effect of the sgRNA treatment on commission errors is not dramatic. An ongoing project in our laboratory is currently addressing how *Dcc* in dopamine neurons guides normative development using male and female mice.

We have now added the following text to the results section to further discuss this important issue, please see page 7:

“While the AUC does not differ significantly between the LacZ and with Dcc-sgRNA saline groups (Figure 5m, inset), their response curves are not visually identical. This is not surprising, considering that conditional genetic downregulation of Dcc levels in DA neurons produces “gene dose-dependent” effects on DA development, with more pronounced changes in

homozygous conditional knockouts than in heterozygotes.^{53,55,60} Forthcoming studies will provide further answers to how the upregulation of Dcc expression impacts normative dopamine development.”

Minor:

The first sentence, “adolescence is a conserved period of life”, is unclear in its meaning—do the authors mean to say it is evolutionarily conserved?

We have now changed the text to explicitly say that:

*“Adolescence is an **evolutionarily** conserved period of life, encompassing the gradual transition from a juvenile to an adult state.”*

References :

Cuesta, S. *et al.* Dopamine Axon Targeting in the Nucleus Accumbens in Adolescence Requires Netrin-1. *Frontiers Cell Dev Biology* 8, 487 (2020).

Cuesta, S. *et al.* DCC-related developmental effects of abused- versus therapeutic-like amphetamine doses in adolescence. *Addict Biol* e12791 (2019) doi:10.1111/adb.12791

Manitt, C. *et al.* The netrin receptor DCC is required in the pubertal organization of mesocortical dopamine circuitry. *J Neurosci* 31, 8381–8394 (2011).

Phillips, R. A. *et al.* An atlas of transcriptionally defined cell populations in the rat ventral tegmental area. *Cell Reports* 39, 110616 (2022).

Reynolds, L. M. *et al.* Quantifying Dopaminergic Innervation in Rodents Using Unbiased Stereology. in *Dopaminergic System Function and Dysfunction: Experimental Approaches* (eds. Evans, J. A. F. & Vargass, P. F. H.) vol. 193 31–63 (Springer, 2022).

Reynolds, L. M. *et al.* Amphetamine in Adolescence Disrupts the Development of Medial Prefrontal Cortex Dopamine Connectivity in a dcc-Dependent Manner. *Neuropsychopharmacol* 40, 1101–1112 (2015).

Reynolds, L. M. *et al.* DCC Receptors Drive Prefrontal Cortex Maturation by Determining Dopamine Axon Targeting in Adolescence. *Biol Psychiat* 83, 181–192 (2018).

Shansky, R. M. Are hormones a “female problem” for animal research? *Science* 364, 825–826 (2019)

REVIEWERS' COMMENTS

Reviewer #1 (Remarks to the Author):

I appreciate the very comprehensive approach to addressing my concerns. The body of work is satisfactory for publication in my opinion.

Reviewer #2 (Remarks to the Author):

The authors have addressed my concerns and the manuscript should be accepted without further revisions.

Reviewer #3 (Remarks to the Author):

The authors have done a terrific job responding to reviewer critiques, and I believe the revised manuscript is improved and will be impactful.

REVIEWERS' COMMENTS

Reviewer #1 (Remarks to the Author):

I appreciate the very comprehensive approach to addressing my concerns. The body of work is satisfactory for publication in my opinion.

We thank the reviewer for their time and their remarks, which have improved the quality of the manuscript.

Reviewer #2 (Remarks to the Author):

The authors have addressed my concerns and the manuscript should be accepted without further revisions.

We thank the reviewer for their time and their remarks, which have improved the quality of the manuscript.

Reviewer #3 (Remarks to the Author):

The authors have done a terrific job responding to reviewer critiques, and I believe the revised manuscript is improved and will be impactful.

We thank the reviewer for their time and their remarks, which have improved the quality of the manuscript.